



**Improved representation of river runoff in Estimating the**
**Circulation and Climate of the Ocean Version 4 (ECCOv4)**
**simulations: implementation, evaluation and impacts to**
**coastal plume regions**
Yang Feng[1,2,3,7], Dimitris Menemenlis[4], Huijie Xue[1,2], Hong Zhang[4], Dustin Carroll[5,4],
Yan Du[1,2], Hui Wu[6]
1. State Key Laboratory of Tropical Oceanography, South China Sea Institute of Oceanology, Chinese
Academy of Science, Guangzhou, China
2. Southern Marine Science and Engineering Guangdong Laboratory, Guangzhou, China
3. Institution of South China Sea Ecology and Environmental Engineering, Chinese Academy of
Sciences, Guangzhou, China
4. Jet Propulsion Laboratory, California Institute of Technology, Pasadena, California, USA
5. Moss Landing Marine Laboratories, San José State University, Moss Landing, California, USA
6. State Key Laboratory of Estuarine and Coastal Research, East China Normal University, Shanghai,
China
7. Guangdong Key Laboratory of Ocean Remote Sensing, South China Sea Institute of Oceanology,
Chinese Academy of Sciences
*Correspondence to*: Yang Feng (yfeng1982@126.com)



**Abstract**
In this study, we improve the representation of global river runoff in the Estimating the
Circulation and Climate of the Ocean Version 4 (ECCOv4) framework, allowing for a more realistic
treatment of coastal plume dynamics. We use a suite of experiments to explore the sensitivity of coastal
plume regions to runoff forcing, model grid resolution, and grid type. The results show that simulated
Sea-Surface Salinity (SSS) is reduced as the model grid resolution increases. Compared to Soil
Moisture Active Passive (SMAP) observations, simulated SSS is closest to SMAP when using Daily,
Point-source Runoff (DPR) and the intermediate resolution LLC270 grid. The Wilmott skill score,
which quantifies agreement between models and observations, yields up to 0.92 for large rivers such as
the Amazon. There was no major difference in SSS for tropical and temperate coastal rivers when the
model grid type was changed from ECCO v4 latitude-longitude-polar cap grid to ECCO2 cube-sphere
grid. We also found that using DPR forcing and increasing model resolution from the coarse resolution
LLC90 grid to the intermediate resolution LLC270 grid elevates the river plume area, volume, and
freshwater transport, along with stabilizing stratification and generally shoaling the mixed layer depth
(MLD). Additionally, we find that the impacts of increasing model resolution from intermediate
resolution LLC270 grid to high resolution LLC540 grid are regionally dependent. The Mississippi
River Plume is more sensitive than others regions, possibly because the wider and shallower Texas-
Louisiana shelf drives a stronger baroclinic effect, as well as relatively weak sub-grid vertical mixing
and adjustment in this region. The results indicate that due to the complex bathymetry and dynamic
behaviour of coastal environments, it may be challenging for spatially-unified resolution models to
capture process-rich fidelity and obtain computational efficiency for coastal interfaces on a global
scale. Our results offer a benchmark for representing Land-Ocean-Aquatic-Continuum (LOAC)
processes in global models and data assimilation products and will help advance predictions of land-
ocean-atmospheric feedbacks seamlessly in the next generation of earth system models.





## 1 Introduction

Coastal plume regions represent a small fraction of Earth's surface, and played as an active component in global cycling of carbon and nutrients (Bourgeois et al., 2016; Carroll et al., 2020; Fennel et al., 2019; Lacroix et al., 2020; Landschützer et al., 2020; Roobaert et al., 2019). Recent satellite-based observations with quasi-global coverage has been greatly improved to monitor Sea Surface Salinity, a key tracer for tracking the river plumes. The European Space Agency (ESA) Soil Measure and Ocean Salinity (SMOS, Mecklenburg et al., 2012) with 33 km/10 day and National Aeronautics and Space Administration (NASA) Soil Moisture Active Passive (SMAP) missions with 40 km/8 day space/time gridding are acquiring SSS observations with sufficient resolution to track the plume pathways and evaluate coastal plume dynamics (Fournier et al. 2016a, b; 2017a,b; 2019; Gierach et al. 2013; Liao et al., 2020). To date, however, the coastal plume regions has not been explicitly resolved in most global Ocean General Circulation Models (OGCMs), Earth System Models (ESMs), and Global Ocean Data Assimilation System (GODAS) products (Ward et al., 2020). As a result, the plume region produced by OGCMs, ESMs, and GODAS are not consistent with satellite observations. For example, Fournier et al. (2016a) found that the 1/12º global circulation Hybrid Coordinate Ocean Model (HYCOM) did not accurately capture SSS during extreme flood events in the northern Gulf of Mexico. Denamiel et al. (2013) found that the Congo River nearshore SSS in global HYCOM was underestimated compared to other regional simulations, even though the models had comparable horizontal grid resolution. Santini and Caporaso (2018) suggested that most CMIP5 models might lack skill in representing the Congo River Basin runoff and SSS in the vicinity of river mouths. Most OGCMs, ESMs, and GODAS products usually had large grid cells; a few cells may encompass the entire plume. As a result, water delivered to the cells are fully mixed and diluted, therefore, cannot accommodate the complex dynamics. Additionally, riverine freshwater input to the ocean is forced in the top model layer over a pre-determined surface area in the vicinity of river mouths with climatological signal, thus the system disturbance by extreme weather events, e.g. floods and droughts, cannot been explicitly resolved (Griffies et al. 2005; Tseng et al. 2016). Finally, virtual salt fluxes have been widely employed, where freshwater affects salinity without a change in mass or volume flux (Bentsen, 2013; Halliwell, 2004; Timmermann et al., 2009; Volodin et al., 2010). The above model configurations may limit the representation of coastal plume region in global scale models.



Estimating the Circulation and Climate of the Ocean (ECCO) is a data assimilating model that uses
observational data to make the best possible estimates of ocean circulation and its role in climate. The
model takes the cube-sphere (ECCO2) to latitude-longitude-polar-cap (ECCOv4) grid for global
application. Like most OGCMs, ESMs, and GODAS products, the current ECCO route riverine
freshwater from land directly to the ocean by taking observed river runoff as seasonal climatology mass
flux over the top of several surface grid cells near the river mouths (Fekete et al. 2002; Stammer et
al.2004). Recent ECCO efforts have been extended to address the global-ocean estimates of $pCO_2$ and
air-sea carbon exchange (Carroll et al. 2020) and model resolution has been promoted as fine as 1-km
globally to investigate mesoscale-to-submesoscale dynamics in the open ocean (Su et al., 2018).
However, current ECCO is lack of representation of coastal interfaces and related feedbacks limiting
their predictability to global climate change, and may further impeding our ability to make informed
resource management decisions. In this study, we here improve the representation of river runoff in the
ECCO and systematically evaluate model performance in reproducing SSS within the vicinity of large
tropical and temperate river mouths. We also investigate the impact of runoff forcing, model grid
resolution, and grid type on coastal dynamics and critical physical properties near the plume regions. The
goal of this work is to provide a comprehensive sensitivity analysis of runoff forcing in multiple
simulations, which will aid in development of global ECCO that more robustly reflecting the Land-
Ocean-Aquatic-Continuum (LOAC).
The paper is organized as follows. Section 2 briefly introduces the ECCO model and the various
runoff forcing methods used in this study. Section 3 provides a comprehensive evaluation of model
sensitivity to horizontal grid resolution and river forcing. Section 4 discusses the sensitivity of plume
properties and coastal stratification. Results are summarized in Section 5.

**2 Methods**

**2.1 ECCO Simulations and Representation of River Runoff**

In this study, we employ the Massachusetts Institute of Technology general circulation model
(MITgcm; Marshall et al., 1997) in a number of model configurations that have been developed for the
ECCO project (Menemenlis et al., 2005; Forget et al., 2015; Zhang et al., 2018). The ECCO MITgcm
configurations that we use herein solve the hydrostatic, Boussinesq equations on either Cubed Sphere
(CS; Adcroft et al., 2004) or Latitude-Longitude-polar-Cap (LLC; Forget et al., 2015) grids. The cubed-



sphere configuration that we use is the so-called CS510 grid, which was developed for the ECCO2 project
(Menemenlis et al., 2008), consists of 6 faces with 510 $\times$ 510 dimension, and has quasi-homogeneous
horizontal grid spacing of 20 km. We also consider three different LLC grid configurations: LLC90,
LLC270, and LLC540, which have, respectively, 1º, 1/3º, and 1/6º nominal horizontal grid spacing. The
LLC grids are aligned with lines of latitude and longitude between 70° S and 57° N, and locally
isotropic with grid spacing varying with latitude. In the tropics, the LLC grid is refined in the meridional
direction to better resolve zonal currents. At high latitudes, the LLC grid is adapted to a two-dimensional
conforming mapping algorithm for spherical geometry. For our experiments, we use LLC# horizontal
grids, where the # is the number of points along one-quarter of the Equator. Therefore, LLC90 means
360 grid points circle the equator. The model has 50 vertical z-levels; vertical resolution is 10 m in the
top 7 levels and telescopes to 450 m at depth. We use a third-order, direct-space-time (DST-3) advection
scheme, while vertical advection uses an implicit third-order upwind scheme. Vertical mixing is
parameterized using the GGL mixing-layer turbulence closure and convective adjustment scheme
(Gaspar et al., 1990). Lateral eddy viscosity in ECCOv4 is harmonic, with a coefficient of $0.005L^2/\Delta t$,
where $L$ is the grid spacing in meters and $\Delta t = 3600s$. Depending on location, the resulting eddy
viscosity varies from $\sim 10^3$ to $\sim 1.6 \times 10^4 \; m^2 s^{-1}$. Additional sources of dissipation in ECCOv4 are
from harmonic vertical viscosity and quadratic bottom drag, along with contributions from the vertical
mixing parameterization. A detailed description of ECCOv4 is provided in Forget et al. (2015).

ECCOv4 uses natural boundary conditions (Huang 1993; Roullet and Madec; 2000), in which

runoff is applied as a real freshwater flux forcing, which allows for material exchanges through the free
surface and more precise tracer conservation compared to virtual salt flux boundary conditions (Campin
et al., 2008). The model uses $z^*$ rescaled height vertical coordinates (Adcroft and Campin, 2004) and
the vector-invariant form of the momentum equation (Adcroft et al., 2004). With $z^*$ coordinates,
variability in free surface height is distributed vertically over all grid cells. For a water column that
extends from the bottom at $z = -H$ to the free surface at $z = \eta$, the $z^*$ vertical coordinate is defined
as $z = \eta + s^* z^*$, where $s^* = 1 + \eta/H$ is the rescaling factor. The Boussinesq, depth-dependent
equations for conservation of volume and salinity under the vector-invariant form of the momentum
equations are:

$$\frac{1}{H}\frac{\partial \eta}{\partial t} + \nabla_{z^*}(s^* v) + \frac{\partial w}{\partial z^*} = s^* F \qquad\qquad (1)$$





$$\frac{\partial (s^*S)}{\partial t} + \nabla_{z^*}(s^*Sv_{res}) + \frac{\partial (Sw_{res})}{\partial z^*} = s^*(D_{\sigma,S} + D_{v,S}) \qquad (2)$$


where F is the surface freshwater flux (includes both precipitation minus evaporation and river runoff),
$\nabla_{z*}$ is the gradient operator on $z^*$ plane. $S$ is the potential salinity, $D_{v,S}$ and $D_{\sigma,S}$ are subgrid-scale
vertical and along iso-neutral mixing, and $v_{res}$ and $w_{res}$ are the horizontal and vertical residual mean
velocity fields. Our daily, point-source runoff (DPR) experiments added freshwater to a single model
grid cell in the first vertical model layer, while the diffuse climatological runoff experiments added it
over multiple horizontal grid cells in the top layer. The amount of freshwater added to each model grid
cell decreased exponentially as a function of distance from river outlets.
**2.2 Sensitivity Experiments**
We first run five experiments, derived from the ECCOv4 set-up, to test the sensitivity of SSS in the
vicinity of large river mouths to ECCOv4 model grid resolution and runoff forcing (Table 1). The LLC90,
LLC270, and LLC540 corresponds to coarse (1° / ~100 km), intermediate (1/3° / ~40 km), and high
(1/6° / ~20 km) resolution from low- to mid-latitudes. LLC90C and LLC270C are forced by monthly
climatological runoff from Fekete et al. (2002). The runoff has a spatial resolution of ~1° and has been
linearly interpolated to each grid cell. Therefore, runoff may be fluxed into a single grid cell in the coarse-
resolution run and over several grid cells in the high-resolution run. The twin experiments, LLC90R and
LLC270R, as well as the highest resolution run LLC540R, use Japanese 55-year atmospheric reanalysis
(JRA55-DO) river forcing dataset (Suzuki et al., 2017; Tsujino et al., 2018). JRA55-DO includes daily
river runoff is generated by running a global hydrodynamic model forced by adjusted land-surface runoff.
Comparing to the Fekete ECCOv4 runoff, JRA55-DO runoff has daily output; therefore, it can resolve
interannual variability and extreme floods and drought events. We add JRA55-DO runoff as point source
flux at a single grid cell adjacent to river outlets. In addition to the LLC grid, two additional experiments
are conducted on the widely-used cube-sphere ECCO2 grid to investigate model sensitivity to the choice
of grid topology (Table 1). CS510C is an ECCO2 run with monthly climatological runoff from Stammer
(2004). The Stammer runoff is spread over a pre-determined surface area in the vicinity of river mouths.
The spreading radius decreases exponentially with a 1000-km e-folding distance. Spatial fields of runoff
forcing for ECCOv4, ECCO2, and JRA55-DO are shown in Figure S1.




| # | Experiment Name | Grid Type | Runoff Forcing | Grid spacing |
|---|---|---|---|---|
| 1 | LLC90C | Lat-Lon-Cap | ECCOv4 Climatology | 55–110 km |
| 2 | LLC90R | Lat-Lon-Cap | JRA55-do | 55–110 km |
| 3 | LLC270C | Lat-Lon-Cap | ECCOv4 Climatology | 18–36 km |
| 4 | LLC270R | Lat-Lon-Cap | JRA55-do | 18–36 km |
| 5 | LLC540R | Lat-Lon-Cap | JRA55-do | 9–18 km |
| 6 | CS510C (Standard ECCO2) | Cube-sphere | ECCO2 Climatology | ~19 km |
| 7 | CS510R | Cube-sphere | JRA55-do | ~19 km |

**Table 1:** Summary of all experiments. The ECCOv4 and ECCO2 climatological runoff is derived from Fekete et al. 2002 and Stammer et al. 2004, respectively. A comparison of runoff forcings is shown in Figure S1.

Each sensitivity experiment is integrated for 26 years (1992–2017) and we analyze the final 3-year period (1 January 2015 to 31 December 2017). We begin our analysis in January 2015 because the high-resolution SMAP observations, which we use to evaluate model skill, are available from 1 April 2015. Ten large rivers at eight coastal regions spanning from low- to mid-latitudes are selected for detailed analysis; these include: the Amazon and Orinoco (AZ); Congo (CG), Changjiang (CJ), Ganges and Brahamptura (GB), Mississippi (MR), Parana (PA), Mekong (MK), and Columbia (CO) rivers **(Figure 1)**.

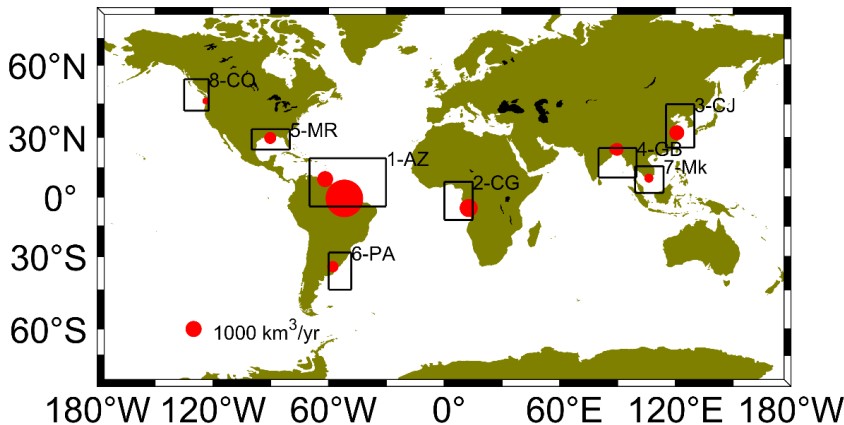

**Figure 1:** The 10 large rivers (red circles) in at 8 coastal regions (white boxes) used in our analysis: Amazon and Orinoco (South America, noted as region 1), Congo (Africa, region 2), Changjiang (Asia, region 3), Ganges and Brahamptura (Asia, region 4), Mississippi (North America, region 5), Parana (South America, region 6), Mekong (Asia, region 7), Columbia (North America, region 8). Red circle size is scaled by the climatological river discharge magnitude.





**2.3 Target Diagram and Willmott Skill Score**


The first part of our study compares the simulated salinity with the synchronized SMAP SSS


observations from 1 April 2015 to 31 December 2017. The level 3 SMAP version-3 SSS was produced


by the Jet Propulsion Laboratory (ftp://podaac-ftp.jpl.nasa.gov Yueh et al., 2013, 2014). We also


compare climatological SSS during this period with the World Ocean Atlas 2018


(https://www.nodc.noaa.gov/OC5/woa18/). For quantitative comparison, we use the Willmott skill score


(Willmott, 1981), a widely-used metric for quantifying agreement between models and observations. The


Willmott score is calculated as:


$$W_{skill} = 1 - \frac{\sum_{i=1}^{n}(M_i - O_i)^2}{\sum_{i=1}^{n}(|M_i - \bar{O}| + |O_i - \bar{O}|)^2} \tag{3}$$


where $M_i$ is the model estimate at $t_i$, $O_i$ is the observation at time $t_i$, $\bar{O}$ is the mean of the observations,


and $n$ is the number of time records for comparison. Specifically, $W_{skill} = 1$ indicates perfect agreement


between model and observations; $W_{skill} = 0$ indicates that the model skill is equivalent to the


observational mean.


Furthermore, we conduct our skill assessment for model SSS on multiple experiments across several


regions. We also use Target diagrams (Jolliff et al. 2009) to efficiently visualize a suite of skill metrics.


Target diagrams are plotted in a Cartesian coordinate system with the x-axis representing the unbiased


root-mean-square-deviation (RMSD), the y-axis represents the bias, and the distance between the origin


and any point within the Cartesian space representing total RMSD.


$$Bias = \frac{\sum_{i=1}^{n}(M_i - O_i)}{n} = \bar{M} - \bar{O} \tag{4}$$


$$Unbiased\ RMSD = \sqrt{\frac{\sum_{i=1}^{n}[(M_i - \bar{M}) - (O_i - \bar{O})]^2}{n}} \tag{5}$$


$$RMSD = \sqrt{\frac{\sum_{i=1}^{n}(M_i - O_i)^2}{n}} \tag{6}$$


$\bar{M}$ represents the mean of the model estimates. These three skill assessment statistics are


particularly useful, as bias reports of the size of the model-observation discrepancies. Bias values near


zero indicate a close match, though it can be misleading as negative and positive discrepancies can cancel


each other. The unbiased RMSD removes the mean and is a pure measure of how model variability differs


from observational variability. The total RMSD provides an overall skill metric, as it includes


components for assessing both the mean (bias) and the variability (unbiased RMSD), i.e.,


$$(Bias)^2 + (Unbiased\ RMSD)^2 = (RMSD)^2 \tag{7}$$






We normalize the bias, unbiased RMSD, and total RMSD by the observational standard deviation ($\sigma_0$)
to allow for the display of multiple experiment and regional SSS observations on a single Target Diagram.
According to the definition of unbiased RMSD, the value should always be positive. However, the X <
0 region of the Cartesian coordinate space may be utilized if the unbiased RMSD is multiplied by the
sign of the standard deviation difference ($\sigma_d$):
$$\sigma_d = sign(\sigma_m - \sigma_o) \quad (11)$$
The resulting target diagram thus provides information about whether the model standard deviation is
larger (X > 0) or smaller (X < 0) than the observation's standard deviation, in addition to if the model
mean is larger (Y > 0) or smaller (Y < 0) than the observation's mean.
**2.4 Definition of plume characteristics**
We investigate the role of grid resolution and runoff forcing using several key metrics: plume area,
volume, and freshwater thickness. The plume area is defined as regions with SSS below a given salinity
threshold $S_A$. The freshwater volume, relative to the reference salinity, $S_0$, is defined as the integral of
the freshwater fraction
$$V_f(S_A) = \iiint_{s<s_A} \frac{S_0 - S(z)}{S_0} dV$$
where the volume integral is bounded by the isohaline $S_A$. Here, we assume the maximum salinity in
each selected region as the reference salinity $S_0$. The freshwater thickness $\delta_{fw}$ represents the equivalent
depth of freshwater and is computed as:

$$\delta_{fw} = \int_{-h}^{\eta} \frac{S_0 - S(z)}{S_0} dz$$

$S(z)$ is the depth-dependent diluted salinity due to the river discharge, $\eta$ is the sea level, and $h$ is the
bottom depth.
We also compute the freshwater flux along arcs (i.e., the edge of river mouth regions shown in
Figure S3). The freshwater flux is defined as:
$$FW_{Trans} = \int_{x_1}^{x_2} \int_{-H}^{\eta} \frac{S_0 - S(z)}{S_0} v_n \, dx dz$$





where $v_n$ is the velocity component normal to the arc (positive outward). $S(z)$ is the depth-dependent
diluted salinity; $dz$ is the layer depth, and $dx$ is the length of the cell. We interpolated the LLC270 and
LLC540 model results to the SMAP grid for unified dx and used the vertical cell depth from the model.
**3. Comparison with Observations**

We first estimate how the various ECCOv4 LLC simulations (Table 1) compare to observations in

the vicinity of 10 large river mouths. The synchronized SMAP SSS (01 April 2015 to 31 December 2017,
33-month) is used as the main verification dataset (Yueh et al., 2013, 2014). SMAP SSS has been
documented to exhibit bias compared to observed SSS in shallow waters near river mouths (Fournier et
al., 2017). Therefore, as an indication of absolute SSS, we also compare the model simulations to the
World Ocean Atlas 2018 (WOA18). We note that there may be relatively few observations incorporated
into the objectively-analysed WOA18 product near the coast, which may over-smooth salinity fronts.
Additionally, WOA18 is a 55-year climatology from 1955–2010; therefore, we can only compare model
climatology from 2015–2017. Overall, we use SMAP and WOA18 as "observational references", where
our model-observation comparisons provide useful information on how SSS changes between
experiments rather than determine which experiment is closer to the real world.

The upper 10-m SSS biases relative to SMAP, averaged over the 33-month period, for CS510C

(standard ECCO2) as well as the LLC540R (highest resolution) are shown in Figure S2. Both SMAP and
WOA18 have 1/4° horizontal grid resolution, therefore, we interpolated all model fields to this grid. For
both simulations, negative biases are found from low- to mid-latitudes, while positive biases occur at
high latitudes. When focusing on large river mouth regions (e.g. AZ, PA and CJ), the SSS bias is reduced
in LLC540R. This demonstrates that the choice of runoff forcing impacts on SSS at predominantly local
scales; however, background currents can transport the signal downstream or offshore to the open ocean
(Liu et al., 2009; Molleri et al., 2010).

Next, we compute mean model SSS near all selected river mouth regions, along with SMAP and

WOA18 (Table 2). The corresponding Willmott Skill (WS) numbers are listed in Table 3. We use the 1st
Empirical Orthogonal Function (EOF) derived from WOA18 to determine river mouth regions, since
WOA18 represents persistent low-salinity zones over the 30-year period. We remove the mean SSS field
before the EOF analysis. The 1st mode explains ~47–67% of the total variance. We then reconstruct the
dominant SSS anomaly field by multiplying the 1st PC with the spatial pattern. Locations with salinity
that is 1~2 PSU lower in the reconstructed SSS field is taken as the river mouth region, and all eight



regions are shown in Figure S3. Near the selected large river mouths, experiments with daily, point-
source runoff forcing generally has lower SSS than experiments forced by climatological runoff
(LLC90R vs. LLC90C; LLC270R vs. LLC270C; CS510R vs CS510C). Increasing model resolution
generally results in regions becoming fresher (Table 2; LLC90C to LLC270C; LLC90R to LLC270R to
LLC540R).

| River Mouth | Abb. | Discharge (m³/yr) | WOA 18 | SMAP | LLC 90C | LLC 90R | LLC 270C | LLC 270R | LLC 540R | CS 510C | CS 510R |
|---|---|---|---|---|---|---|---|---|---|---|---|
| Amazon / Orinoco | AZ | 6440 | 32.7 | 27.5 | 34.0 | 34.1 | 31.7 | 28.2 | 24.6 | 34.3 | 23.8 |
| Congo | CG | 1270 | 33.6 | 33.7 | 34.7 | 34.3 | 34.6 | 33.9 | 33.7 | 34.9 | 34.1 |
| Changjiang | CJ | 907 | 32.9 | 31.4 | 33.1 | 32.8 | 33.0 | 32.2 | 31.8 | 32.5 | 30.9 |
| Ganges / Brahamptura | GB | 643 | 29.3 | 27.5 | 30.9 | 29.4 | 29.5 | 27.2 | 23.9 | 29.7 | 25.6 |
| Mississippi | MR | 552 | 33.5 | 34.8 | 35.8 | 34.7 | 35.8 | 34.1 | 33.8 | 35.3 | 34.1 |
| Parana | PA | 517 | 28.9 | 27.3 | 33.7 | 31.0 | 31.1 | 24.7 | 20.0 | 33.8 | 20.0 |
| Mekong | MK | 504 | 32.9 | 32.9 | 33.5 | 32.6 | 32.3 | 30.3 | 31.0 | 31.8 | 28.5 |
| Columbia | CO | 167 | 30.7 | 31.0 | 32.0 | 31.7 | 31.7 | 30.8 | 30.3 | 31.4 | 30.4 |

**Table 2:** The SSS near river mouth for WOA18, SMAP, and all experiments for the selected regions

An SSS comparison between SMAP and WOA18 showed no consistent patterns. Explaining
differences between the two datasets is beyond the scope of this study; however, we acknowledge that
some large differences complicate our comparison. Use of DPR forcing yielded higher $W_{skill}$ scores in
most regions when using SMAP as the observational reference, but not when using WOA18 (e.g.
LLC270C/R for AZ, GB, MK). Additionally, a comparison to SMAP shows that $W_{skill}$ scores become
higher when model resolution increases from 1° to 1/3° (LLC90C vs LLC270C, LLC90R vs LLC270R),
but lower when further increases to 1/6° (e.g. LLC270540R for Amazon, Ganges/Brahmaputra, Parana).
In contrast, $W_{skill}$ lacks consistency when comparing to WOA18 (e.g. at GB, $W_{skill}$ associated with
LLC90R exceeds that of both LLC270R and LLC540R). The higher or lower $W_{skill}$ score is consistent
with how much the model deviates from the observational reference. At the AZ region, SSS from
LLC270R is less than 1 psu lower than the SMAP average, while LLC540R is roughly 3 psu lower.
Therefore, LLC270R receives a skill score 0.92, higher than LLC540R (0.74). This also occurs with
LLC270C and LLC270R when using WOA18 as the reference. For rivers in tropical and temperate zones,





the CS510 grid has a resolution comparable with the LLC540 grid, therefore, the SSS and skill scores
are comparable between CS510R and LLC540R. Since the model grid type has a negligible impact on
SSS for low- to mid-latitude rivers, we next focus on model sensitivity to grid resolutions.

| River Mouth | LLC90C | LLC90R | LLC270C | LLC270R | LLC540R | CS510C | CS510R |
|---|---|---|---|---|---|---|---|
| with SMAP | | | | | | | |
| Amazon / Orinoco | 0.50 | 0.50 | 0.71 | 0.92 | 0.79 | 0.46 | 0.73 |
| Congo | 0.58 | 0.64 | 0.69 | 0.89 | 0.88 | 0.60 | 0.87 |
| Changjiang | 0.53 | 0.59 | 0.51 | 0.64 | 0.83 | 0.59 | 0.85 |
| Ganges / Brahamptura | 0.61 | 0.71 | 0.69 | 0.85 | 0.69 | 0.57 | 0.70 |
| Mississippi | 0.55 | 0.79 | 0.53 | 0.77 | 0.75 | 0.49 | 0.72 |
| Parana | 0.37 | 0.51 | 0.45 | 0.62 | 0.40 | 0.37 | 0.40 |
| Mekong | 0.79 | 0.90 | 0.77 | 0.54 | 0.63 | 0.74 | 0.38 |
| Columbia | 0.46 | 0.60 | 0.49 | 0.73 | 0.74 | 0.27 | 0.61 |
| with WOA | | | | | | | |
| Amazon / Orinoco | 0.73 | 0.69 | 0.87 | 0.64 | 0.47 | 0.54 | 0.44 |
| Congo | 0.60 | 0.67 | 0.70 | 0.94 | 0.95 | 0.64 | 0.92 |
| Changjiang | 0.82 | 0.94 | 0.68 | 0.70 | 0.72 | 0.78 | 0.54 |
| Ganges / Brahamptura | 0.72 | 0.90 | 0.92 | 0.78 | 0.51 | 0.73 | 0.59 |
| Mississippi | 0.46 | 0.59 | 0.46 | 0.68 | 0.73 | 0.49 | 0.66 |
| Parana | 0.40 | 0.45 | 0.45 | 0.42 | 0.29 | 0.40 | 0.29 |
| Mekong | 0.84 | 0.87 | 0.83 | 0.45 | 0.54 | 0.71 | 0.30 |
| Columbia | 0.51 | 0.62 | 0.63 | 0.87 | 0.84 | 0.51 | 0.90 |

**Table 3:** The Willmott skill score for each run as compared with WOA18 and SMAP. The river mouth
was recognized by the 1st EOF of WOA18 (See Figures 5 and S1). Note that WOA18 data are a 30- year
climatology (1981—2010) and not in the same period as SMAP and experiments.

To better compare the sensitivity of SSS to river forcing, we provide zoomed-in plots of the same
comparison shown in Figure 1 for AZ, MR, and CO Rivers for all LLC simulations, representing large,
medium, and small rivers (Figure 2). The positive bias is greatly reduced when applying daily, point-
source river forcing, as well as increasing the horizontal grid resolution.



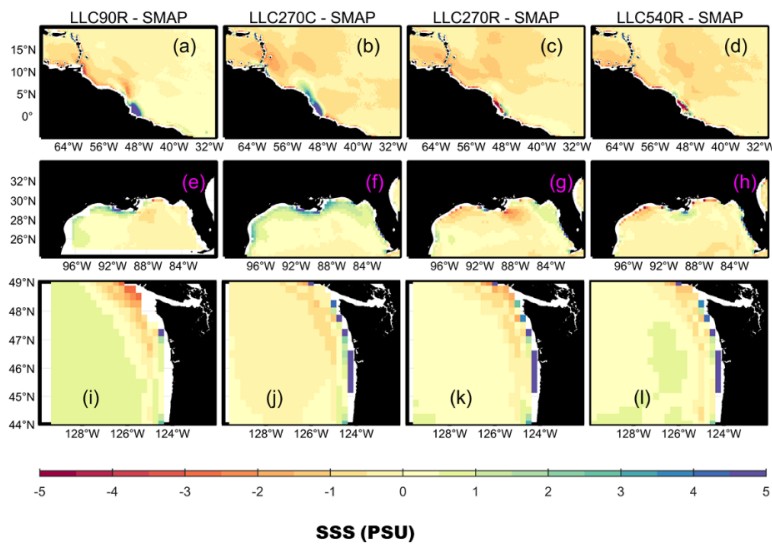


**Figure 2:** Zoomed-in view of SSS difference between model experiments and SMAP observations for
large (Amazon, a–d), medium (Mississippi, e–h), and small (Columbia, i–l) rivers.

Timeseries for all LLC simulations and SMAP, at these three river mouths, are shown in **Figure 3**.
As in Figure 2, the bias decreases when daily, point-source river forcing is used and as the horizontal
grid resolution increases. Additionally, there is substantial improvement in reproducing observed SSS
variability. When using climatological runoff forcing, only seasonal variability is resolved in the
simulations. When using DPR forcing, both the observed seasonal and interannual SSS variability are
reproduced by the model. For example, the 2017 abnormally low SSS near the Amazon river mouth is
associated with an extreme flooding event (Barichivich et al., 2018).

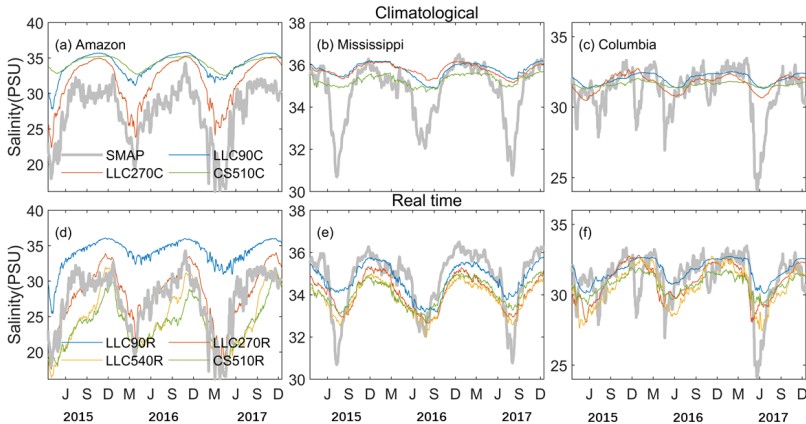






**Figure 3:** Areal averaged SSS in the Amazon, Mississippi and Columbia river mouth regions (see Figure
S3) with climatological and daily, point-source runoff forcing for SMAP (thick gray line) and
experiments (thin colored lines) with varying horizontal grid resolution. The method used to characterize
the river mouth region is described in Section 3.
Next, we quantify the difference in mean and variance between the SSS time series of LLC
simulations and that of SMAP under different runoff forcing scenarios **(Figure 4)**. The experiment with
intermediate resolution shows that both the normalized bias and unbiased RMSD with daily forcing is
lower than experiments with climatological forcing. Furthermore, using daily point-source runoff forcing
results in mean simulated SSS that is closer to mean SMAP SSS in many regions (e.g. AZ, CG, GB, and
CO). The total normalized RMSD improvements are primarily due to the normalized bias decrease. We
find that most unbiased RMSD remains negative when varying the runoff forcing from climatological to
daily. This implies that the variance of LLC simulations remains lower than SMAP observations as the
runoff forcing changes. The only two exceptions are Congo and Mekong, possibly because JRA55DO
runoff has stronger variability compared to the ECCOv4 climatology for the two rivers.

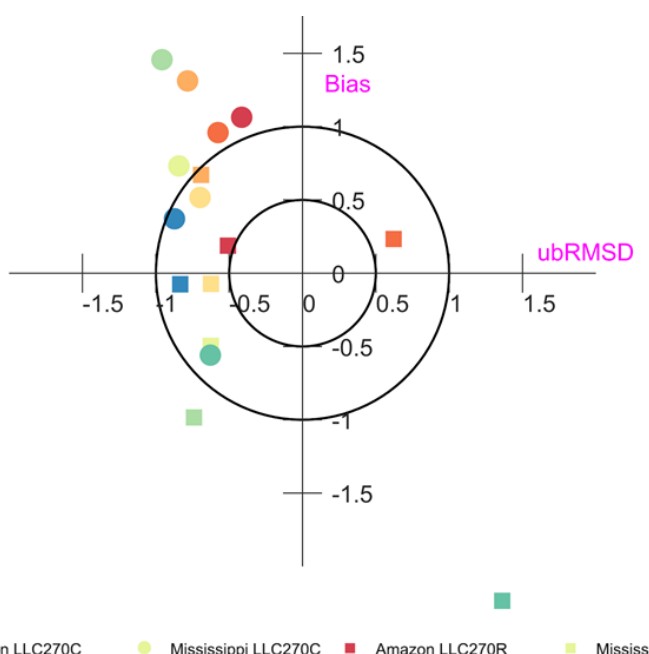

**Figure 4:** SSS target diagram near the selected river mouths (see Figure S3) for LLC270C and LLC270R
simulations.



We also examine the bias and variance on the Target Diagram for experiments with varying
grid resolution but similar daily runoff forcing **(Figure 5)**. Our results show that the normalized bias
decreases as the model resolution increases,while the unbiased RMSD decreases slightly with the sign
remaining negative as the model resolution increases. This occurs everywhere, except for the two largest
rivers (AZ and CG) where the sign becomes positive for LLC540R, indicating that the model variance
exceeds the SMAP variance when using the high-resolution grid. In summary, the comparison with
synchronized SMAP shows that using daily runoff and finer horizontal grid resolution improves the
representation of SSS variability but at a cost of increased SSS bias.

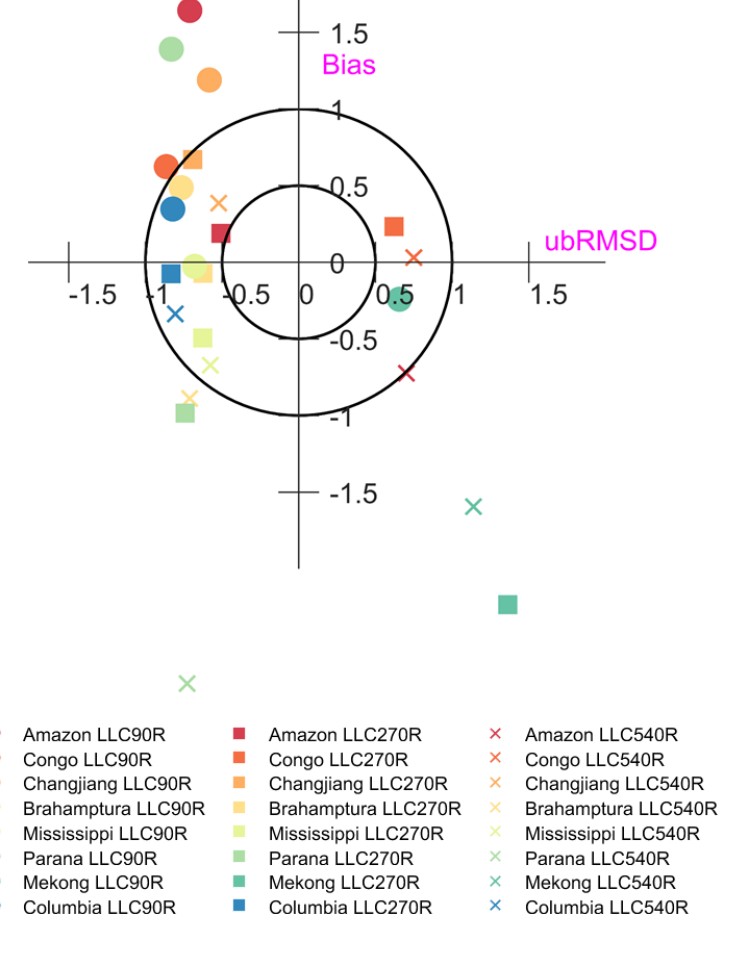

**Figure 5:** Same as Fig. 5, but for LLC90R, LLC270R, and LLC540R.

## 4. Impact on River Plume Properties

### 4.1 EOF analysis of SSS

We next investigate how model runoff improvements impact river plume properties such as plume

area, volume, and freshwater thickness. We first evaluate the plume SSS signature and dynamics through

EOFs; the mean is removed before the EOF analysis. The first and second mode of AZ, MR, and CO

using the same grid resolution but different runoff forcing is shown in Figures 6 and S4. The spatial

pattern reveals the salinity anomaly caused by the runoff, while the PC timeseries shows the timing. A

single value in spatial pattern or PC timeseries doesn't have a clear physical meaning, but together they

reveal how much salinity deviates from the mean. The PC timeseries for experiments with DPR forcing

clearly shows similar seasonal cycles, albeit with larger amplitudes and interannual variability.

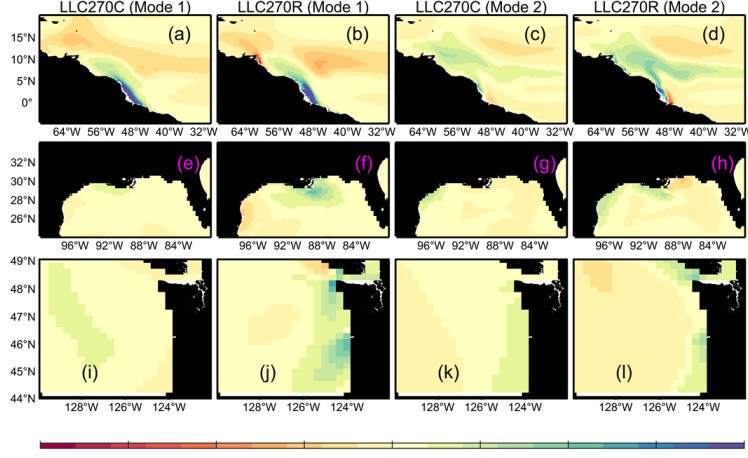

**Figure 6:** 1$^{st}$ and 2$^{nd}$ EOF spatial patterns from the LLC270C and LLC270R simulations for the Amazon, Mississippi, and Columbia rivers. The corresponding PC timeseries are normalized by the standard deviation and multiplied back to the spatial mode.





For the AZ region, the LLC270R and LLC270C spatial patterns are similar for the first and second
mode. The first mode accounts for 59% (72%) of the total variance in LLC270R (LLC270C). The spatial
pattern reveals the presence of a low-salinity tongue; this is located in a narrow band along the
northeastern South American coast from February–June, which is associated with the large river
discharge and the northward-flowing Brazil Current. The second mode of LLC270R (LLC270C)
accounts for 33% (23%) of the total variance. The spatial pattern shows that the plume-like features
extend northwestward to the Caribbean Sea and Central Equatorial Atlantic Ocean from May–September.
This pattern is driven by Ekman currents associated with northeasterly wind stress, and the transport to
the Central Equator is due to the North Equatorial Counter Current (NECC, Lentz 1995a,b).
For the MR region, the first and second mode of LLC270R (LLC270C) explains 53% (66%) and
29% (18%) of the total variance, respectively. The spatial pattern of the first mode is generally similar.
There is a bulge-like plume feature that occupies a region near the MR mouth with a southeast extension
to the central Gulf of Mexico from May–October (Figure S4), while the freshwater signal in the vicinity
of the southeast MR mouth is stronger in LLC270R. The extension of low-salinity waters is due to the
upwelling favorable winds (southwesterly) from late-spring to summer, which transport the MR
freshwater offshore (Walker, 1996).
The first and second mode explains 63% (56%) and 29% (33%) of the variability at the CO region
in LLC270C (LLC270R), respectively. It has been previously recognized that the CO plume exhibits
seasonal variability forced by wind and freshwater discharge (*García Berdeal et al.*, 2002). During winter,
Ekman transport resulting from the northward winds constrains the plume against the Washington coast.
Downwelling-favorable wind stresses strengthens the anti-cyclonic rotation of the river plume, resulting
in a coastally-attached winter plume. In contrast, prevailing southward wind stress results in offshore



Ekman transport; this advects the plume offshore, where it is influenced by the California Current over
long timescales and subsequently veers southward and offshore (Banas et al., 2009). This seasonal
pattern is shown in the first LLC270R mode and second LLC270C mode.

The first and second EOF modes for AZ, MR, and CO with daily runoff forcing in coarse (LLC90R)

and fine (LLC540R) grid resolution are shown in Figures 7 and S5. The plume-like features and
associated dynamics are similar to LLC270 in both runs. Additionally, the higher-resolution LLC540R
resolves fine-scale plume structure for a number of major rivers, which was previously revealed by
satellite observations, regional simulations, or neural network methods (e.g. meanders and rings of the
AZ plume due to the NBC retroflection, Molleri et al. 2010);"horseshoe" patterns of the MR plume
associated with Texas floods (Fournier et al. 2016), and the bidirectional CO plume during variable
summer wind patterns (Liu et al. 2009). Overall, EOF SSS analysis shows that general plume pattern and
dynamics are grid independent; however, fine-scale plume structures are only resolved by high-
resolution simulations.

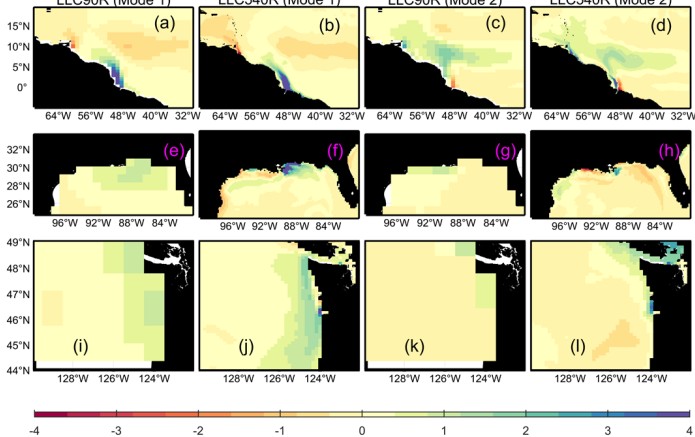

**Figure 7:** Same as Fig. 6 but for the LLC90R and LLC540R simulations. The corresponding PC
timeseries are normalized by the standard deviation and multiplied back to the spatial mode shown in
Figure S5.



**4.2 Plume Area, Volume, Freshwater Thickness, and Transport**

We first calculated plume area using a salinity threshold from 28 to 36 PSU (Figure S6). The

absolute plume area varied with the choice of $S_A$. Since the seasonal and interannual variability are
comparable once the plume can be clearly resolved at the given threshold, we only show the plume area
within $S_A = 30$ psu under the climatological and daily runoff forcing at the coarse, intermediate, and
high resolution runs (Figure 8). Figure 9 presents a time series of freshwater volume within the given
salinity during this period. There is a stronger interannual variability when using DPR, along with larger
plume area and volume during flood years. The MR and CO plume area/volume cannot be explicitly
resolved at the $S_A = 30$ threshold when using the climatological and runoff forcing since river runoff
has been distributed over a broad spatial grids and surface salinity decrease is small. For the AZ region,
the averaged plume area (volume) approaches $6 \times 10^4$ km$^2$ ($7 \times 10^2$ km$^3$) in LLC270C, whereas it is
only about $3 \times 10^4$ km$^2$ ($5 \times 10^2$ km$^3$) in LLC90C. In contrast, the MR and CO plume area (volume)
is easily recognized when using DPR forcing. The AZ plume increases as the grid resolution increases,
reaching 10, 13, and $17 \times 10^4$ km$^2$ in LLC90R, LLC270R, and LLC540R, respectively. The freshwater
volume in coarse-, intermediate-, and high-resolution runs is comparable, with values of ~ 1.5 - 2 $\times 10^2$
km$^3$. The plume area and volume in the MR region is more sensitive to the model grid resolution than
AZ and CO. The LLC540R plume area is ~3–4 times higher than LLC270R, while LLC270R is ~6–7
times higher than LLC90R.  For the CO region, the plume area when using DPR forcing is similar
between intermediate- and fine-resolution experiments, with the area in LLC270R and LLC540R
increasing to ~1 $\times 10^5$ km$^2$ during the 2015 flood year. In contrast, LLC540R maintains a larger plume
volume than the intermediate resolution run.



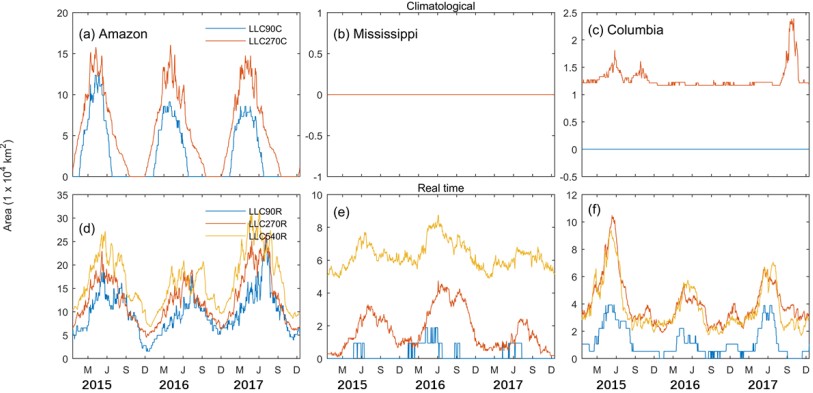

**Figure 8:** Same as Fig. 7 but for the LLC90R and LLC540R simulations. The corresponding PC timeseries are normalized by the standard deviation and multiplied back to the spatial mode shown in Figure S5.

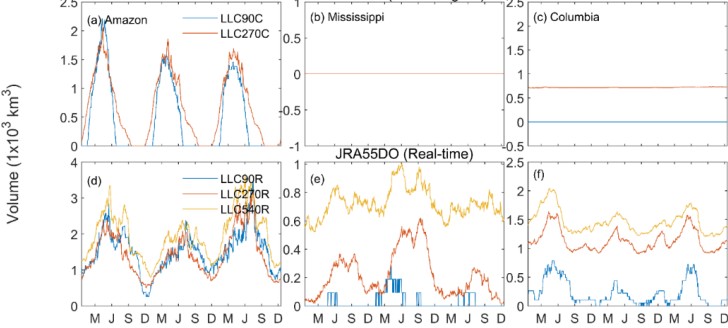

**Figure 9:** Freshwater volume within 30 PSU for the Amazon, Mississippi, and Columbia river regions for various experiments, reference salinity is 36 PSU.

The sensitivity of plume area and volume to runoff forcing and grid resolution reflects the

experiments ability to resolve the horizontal advection and downward mixing of riverine freshwater. This

can be partially reflected in the freshwater thickness calculation, which is shown in Figure 10. For the

intermediate resolution experiments, the maximum freshwater thickness $\delta_{fw}$ is over 10 m, 5 m, and 4 m

near the AZ, MR, and CO river mouths when using DPR, as opposed to 4 m, 2 m, and 2 m, when using

climatological runoff. Additionally, the freshwater thickness in experiments with DPR but different grid

resolutions (LLC270R and LLC540R) demonstrates that a coherent plume rotates and responses to





external wind and background flows; this coastal plume structure is largely absent in the coarser LLC90R.
The coarse resolution experiment exhibits a more diffuse response, with low salinity near MR and CO
river mouths. Note that the runoff forcing is identical between LLC90R, LLC270R, and LLC540R
experiments, and differences in plume area, volume, and freshwater thickness are due to model resolution
alone. The freshwater flux in the higher resolution experiments can result in larger inflow velocities, a
stronger baroclinic response, and consequently a more vigorous coastal plume. The plume area and
volume in MR region is more sensitive to grid resolution — this possibly results from the representation
of shelf bathymetry. The Texas-Louisiana shelf is wider and shoals more gradually from the coastline
compared to the northern Brazilian shelf (AZ) and Washington shelf (CO). When adding the same
amount of freshwater in shallow water regions, high resolution experiments generate a larger pressure
gradient force than the intermediate resolution, which drive a stronger baroclinic effect and elevate
coastal currents. The alongshore currents can advect the MR Plume water downcoast, enlarging the
plume area. Besides, the plume waters may be entrained downward by strong sub-grid vertical mixing
and adjustment, e.g. meso-scale eddies, when flowing offshore to the open ocean as the horizontal
resolution increases from intermediate to high. The eddies in high resolution run at AZ and CR may
break up the plumes, shrinking the area comparing to low-to-intermediate resolution run (Figure S6).





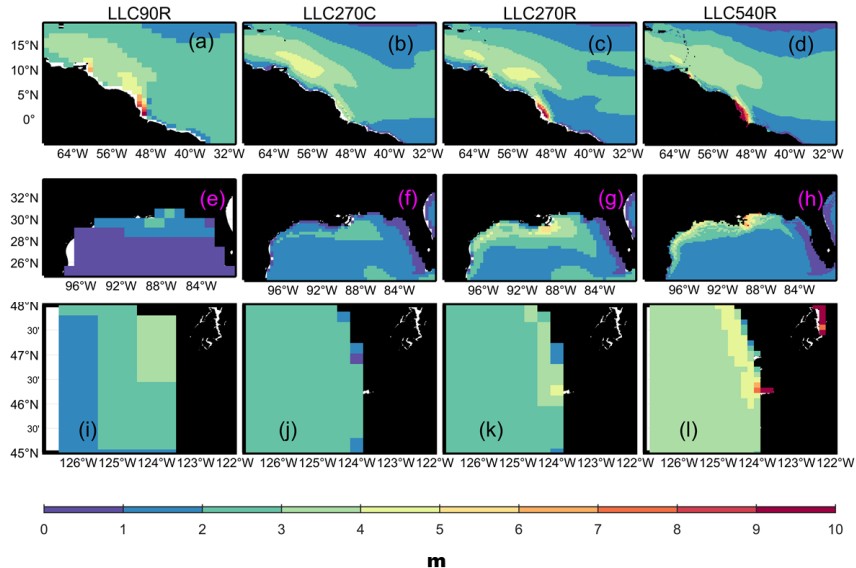

**Figure 10:** Freshwater thickness for LLC90R, LLC270C, LLC270R, and LLC540R experiments.

The integrated freshwater transport is shown in Figure 11. The coarse-resolution LLC90 simulation

does not resolve the selected arc and therefore is not shown. For the LLC270 simulation, the integrated

freshwater transport when using the climatological runoff is smaller than when using DPR in the AZ,

MR, and CO regions. When further increasing model resolution to LLC540, $FW_{Trans}$ only increases at

the MR region, but not in the AZ and CO regions, which is consistent with the plume area/volume

response. The larger freshwater transport at the MR region is mainly due to a strengthening of the coastal

current along the shallow continental shelf. We note that these results may vary depending on the choice

of arc. Additionally, further increases in the model grid resolution may distort the plume characteristic

and lead to different conclusions. Again, this similarity suggests that the baroclinic current becomes

stronger as model grid resolution increases in shallower and wider shelf regions.



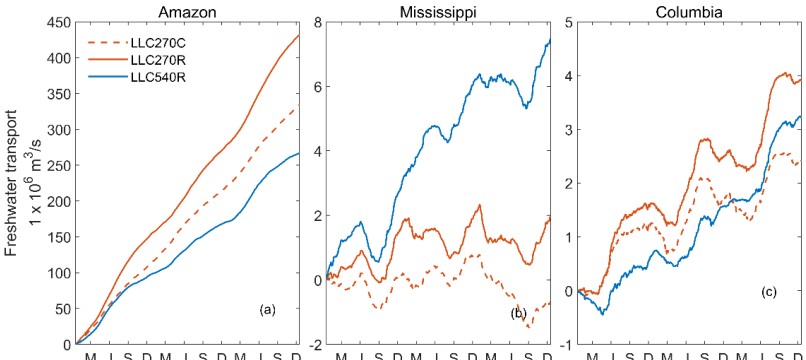


**Figure 11:** Integrated freshwater transport from 2015–2017 along the arc of the Amazon, Mississippi,
and Columbia river mouth. Arc is defined as the edge of the river mouth region shown in Figure S3.


**4.3 Impact on Ocean Properties Associated with SSS**

In this section, we examine the sensitivity of stratification and mixed layer depth (MLD) between
different experiments. Figure 12 shows 3-year averaged vertical profiles of salinity, temperature, vertical
density gradient $d\rho/dz$ ( $\rho$ is the potential density) near the AZ (top), MR (middle), and CO (bottom)
river mouths, respectively. The profiles are averaged over the horizontal regions shown in Figure S3.
The vertical density gradient is an important indicator of stratification strength. The salinity differences
between climatological (LLC90/270C) and DPR forcing (LLC90/270R) are large near the surface and
diminish with increasing depth. The temperature difference when using the two type of runoff forcing is
insignificant, demonstrating that the stratification is primarily determined by salinity and the addition of
freshwater. Additionally, DPR forcing greatly increases subsurface stratification, which implies a
decrease in vertical mixing.
Figure 12 also shows sensitivity of upper-ocean stratification to various model grid resolutions.
The profiles show a significant decrease in salinity from the surface to 50-m depth as the resolution



increases, which impacts the stratification (rightmost panels). We note that the vertical density gradient
has a subsurface maximum in the coarse- and intermediate-resolution run, while the high-resolution
experiment has a surface maximum due to low-salinity water concentrated in the surface level. SST is
highest in LL540R at AZ and MR, reflecting that heat is preserved at the surface due to the increase in
subsurface stratification.

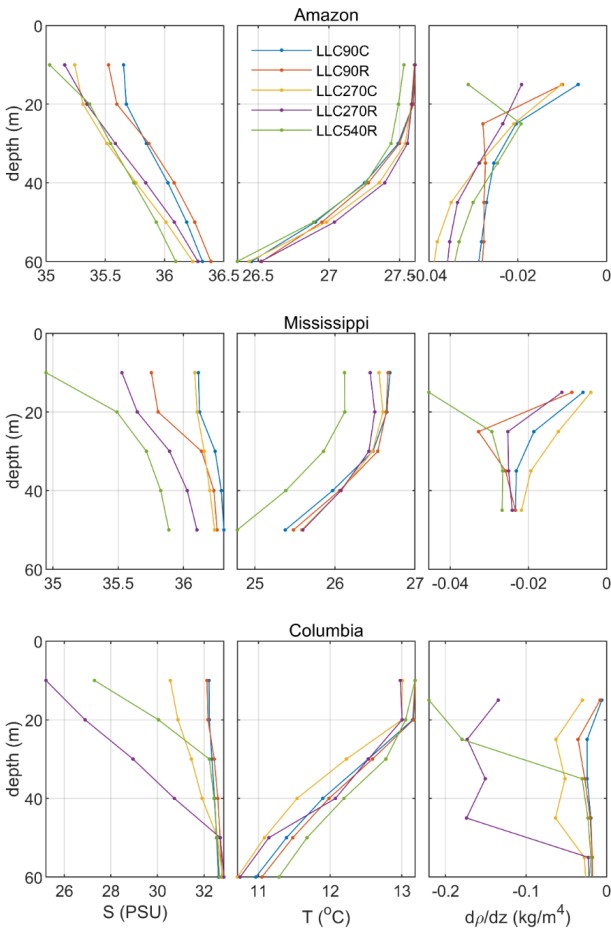


**Figure 12:** Mean 3-year (2015-2017) salinity, potential temperature, and vertical density gradient
($d\rho/dz$) profiles.



The sensitivity of stratification in the above analysis implies that the MLD can be altered by river
forcing and grid resolution. We compare the MLD in the vicinity of AZ, MR, and CO during the
simulation period in Figure 13. The MLD in our calculation uses the threshold method, in which deeper
levels are examined until one is found with density differing from the near surface by more than 0.03
$kg/m^3$ (de Boyer Montégut et al., 2004). This reflects the maximum depth of the boundary layer that is
sustained by riverine freshwater. Interestingly, all experiments simulate the annual cycle of MLD. There
was relatively shallow MLD from April to December, which corresponds to periods of high river
discharge. The MLD in the DPR forcing and high-resolution scenario is shallower than the climatological,
low-resolution scenario, which is consistent with vertical salinity and stratification profiles shown in
Figure 11.

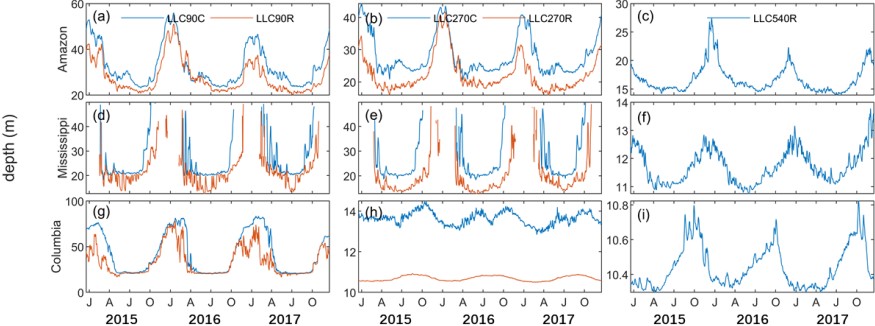

**Figure 13.** 2015–2017 daily MLD, averaged in the vicinity of the Amazon River (upper panels),
Mississippi River (middle panels), and Columbia River (lower panels) mouth. MLD is computed based
on de Boyer Montegut et al. (2004).

**5. Discussion and conclusions**
In this study, we investigate the model sensitivity of runoff forcing and grid resolution and type
under the ECCO framework. We find that DPR greatly improves model representation of global rivers,
with horizontal model resolution having a substantial control on SSS in the vicinity of river mouths. We





observe no major changes in tropical and temperate river mouth SSS when using cube sphere grid or
LLC grid types when using the same river forcing. A comparison with synchronized SMAP observations
shows that the use of DPR forcing and intermediate grid resolution can increase the model performance
in simulating SSS in the vicinity of river mouths. However, further increasing model grid resolution from
intermediate to high may result in an additional SSS bias towards fresher values.

Previous theoretical modelling studies have demonstrated that, in the absence of external forcing,

large river plumes influenced by rotational effects tend to veer anticyclonically and form a bulge region
near the river mouth as well as an along-shore downstream coastal current as Kelvin waves (Kourafalou
et al. 1996; Yankovsky and Chapman, 1997). Additionally, idealized numerical simulations have
revealed that river plume behaviour is greatly impacted by external forcing. Chao (1988a, b)
demonstrated that vertical mixing, bottom drag, and bottom slope greatly impact the spin-up,
maintenance, and dissipation of river plumes. Fong and Geyer (2001) revealed that a surface-trapped
river plume would thin and be advected offshore by cross-shore Ekman transport. Fong and Geyer (2002)
suggested that the ambient current, which is in the same direction as the geostrophic coastal current, can
augment plume transport. Our ECCO experiments are examining the above river plume dynamic theory
globally with realistic topography and external atmospheric forcing. Our EOF analysis of SSS at the AZ,
MR, and CO shelves show that the general spatial and temporal patterns plume related to river discharge,
wind, and currents are independent of the grid resolutions and forcing formulations examined in this
study, which are quite consistent with those previous studies. However, higher resolution and DPR
forcing may be particularly important for resolving the fine-scale plume dynamics for small rivers. Using
DPR forcing and increasing the model grid resolution from coarse to intermediate increases the river
plume area, volume, and freshwater transport, while further increasing the model resolution from





intermediate to high has mostly regional effects. Shallow and wide shelf regions, such as the Mississippi
Delta, are more sensitive compared to AZ and CO. This suggests high resolution model may take more
computational resources without promoting the spatiotemporal scales of interest system dynamics. Two
potential solutions are applying domain nesting scheme and spatially variable model adapting the global-
scale open ocean (~ km) and small-scale coastal and estuarine environments (~ m).

We also found that using DPR forcing and increasing the model grid resolution can stabilize the

water column at subsurface and shoal the MLD. This may have significant implications for
biogeochemical cycles and air-sea exchange in coastal zones. From the biogeochemistry perspective,
freshwater introduced by the river increase shelf stratification, preventing the reoxygenation of bottom
waters and thus may generate large hypoxic regions (Fennel, et al. 2011; Feng et al. 2019). From the air-
sea interaction perspective, on one hand, SST can trigger deep atmospheric convection and strong rainfall.
On the other hand, strong near-surface stratification may inhibit cooling and intensify tropical cyclones
(Cione and Uhlhorn, 2003; Neetu et al., 2012; Rao and Sivakumar, 2003; Sengupta et al., 2008; Vialard
& Allison et al., 2000; Vinaychandran et al., 2002). We envision that future work investigating river
impacts on ocean-atmospheric and earth system dynamics could be accomplished by coupling our
improved ECCO simulations with an atmospheric general circulation model (AGCM).

In the state-of-art OGCMs, ESMs, and most GODAS products, river runoff is incorporated on coarse

resolution grids as augmented precipitation. Climatological runoff forcing is often used in conjunction
with artificial spreading, along with a virtual salt flux scheme. Tseng et al. (2016) examined model
sensitivity to the spreading radius, turbulent mixing parameterization, reference salinity, and vertical
distribution of riverine freshwater on 1° resolution in Community Earth System Model (CESM). For all
factors examined, they found that the model results are most sensitive to the spreading radius, which





substantiates the importance of our finding that the associated plume properties including the area hence
the SSS near river mouths exhibit strong responses when switching the runoff flux from diffusive
climatological to daily point-source.

The present state-of-the art regional scale estuarine models can simulate estuarine hydrodynamics

and biogeochemical processes in a robust manner. The inlet approach, which defines a rectangular breach
in coastal land cells with uniform density and discharge, is widely used (Herzfeld, 2015; Garvine, 2001).
An additional barotropic pressure term may be added to account for pressure gradients induced by the
freshwater plume (Schiller and Kourafalou, 2011). The inlet approach has also been used in global z-
coordinate models by injecting freshwater in multiple vertical grid cells (Griffies et al. 2005). In our
simulations, changes in sea level are redistributed over all vertical grid cells by the rescaled height
vertical coordinate. This is similar to the inlet approach in the regional models, which add a mass or
volume flux of freshwater to a breach in coastal land cells (Garvine 1999). Herzfeld (2015) investigated
the role of model resolution on plume response at the Great Barrier Reef (GBR) using the Regional
Ocean Modeling System (ROMS). The study found that the plume veered left and followed a northward
trajectory to Cape Bowling Green in a 1 km resolution model but not in a 4 km resolution model. Our
findings are consistent with this result; plume properties in our intermediate resolution simulations are
more clearly detected than in the coarse resolution simulations. In addition, our results expand on their
findings by showing that the sensitivity of plume properties and freshwater transport in high resolution
model are highly-dependent on shelf bathymetry. Schiller and Kourafalou (2010) investigated the
dynamics of large-scale river plumes in idealized numerical experiments using HYCOM to address how
the development and structure of a buoyant plume is affected by the vertical and horizontal redistribution
of river inflow and bottom topography. Their experiments show that a narrow inlet, flat bottom facilitates





a larger right-turning plume bulge region compared to a wide inlet, slope bottom (see their Figure 8
Riv2c-f; Riv2c-s). This is complementary to our findings that the MR plume, located on the wide and
shallow LA shelf, has a larger horizontal plume area compared to the AZ and CO plume when increasing
the horizontal resolution from intermediate to high. However, their discussion was limited to an idealized,
rectangular model domain without external forcing, while our model simulations provide a realistic
application to natural river plume systems. The global application of the regional inlet representation of
river forcing was also used by NOAA's Geophysical Fluid Dynamics Laboratory (GFDL) models
(Griffies et al., 2005). An internal, pre-computed salinity source term was introduced into multiple
vertical layers. However, river representation was done through VSF rather than through real mass or
freshwater volume flux. Adding the real volume and mass of freshwater through multiple layers has been
widely used in regional models like ROMS; it may be useful to adapt this technology into future ECCO
simulations and comparing the results with the current surface injection methods.
For the global ocean, river runoff is much smaller than the precipitation and evaporation flux;
therefore, for most OGCMs, ESMs, and GODAS products it is parameterized. One of the most significant
expected signatures of global warming is an acceleration of the terrestrial hydrological cycle (IPCCs,
2019; Piecuch and Wadehra, 2020). Both can significantly affect the magnitude, distribution, and timing
of global runoff, leading to extremes in the frequency and magnitude of floods and droughts. When
considering issues related to water resource management under climate and land use/land cover change,
key question such as "how will coastal oceans be impacted from flood and drought events?" is
challenging to answer (Fournier et al. 2019). In the future, high-resolution global-ocean circulation
models with DPR forcing may help identify the primary forcing mechanisms (such as those from climate-





driven extreme events) that drive spatiotemporal variability of large river plume systems just as skillfully
as regional model set-ups.

Improved model representation of rivers may not be as important for global or basin- scale

hydrological cycles as precipitation and evaporation (Du and Zhang, 2015), but may be critical for the
global carbon cycle (Friedlingstein et al., 2019; Resplandy et al. 2018). River delivers large amounts of
anthropogenic nutrients to the coastal zone (Seitzinger et al., 2005, 2010). The autochthonous production
will transform inorganic nutrients to organic while sequestering atmospheric $CO_2$. More importantly,
rivers also deliver dissolved organic carbon (DOC) and particulate organic carbon (POC) to the coastal
ocean, which can be remineralized and released as $CO_2$ to the atmosphere. Until recently, most global-
ocean biogeochemistry models omitted or poorly represented riverine point sources of nutrients and
carbon. Lacroix et al. 2020 added yearly-constant riverine loads to the ocean surface layer on coarse
resolution (1.5°) model and assessed that $CO_2$ outgassing from river loads accounted for ~10% of the
global ocean $CO_2$ sink. We anticipate that the implementation of DPR forcing and higher-resolution grids
in ESMs and the ECCO biogeochemical state estimates (ECCO-Darwin, Carroll et al., 2020) will help
better resolve the global carbon budget (Friedlingstein et al., 2019).

LOAC development has historically had a low priority in OGCMs, ESMs, and GODAPs and

exchange of freshwater between rivers/estuaries and the coastal ocean has been previously neglected.
Our results demonstrate that the representation of runoff forcing in ECCO simulations is a major source
of bias for coastal SSS. We believed our   improvements of river runoff in ECCO will directly contribute
to: (i) the evaluation, understanding, and improvement of river-dominated coastal margins in global-
ocean circulation models, (ii) investigation of mechanisms that drive seasonal and interannual variability
in coastal plume processes, and (iii) bridging the gap between land-ocean interactions. These efforts will



ultimately help to better resolve land-ocean-atmosphere processes and feedbacks in next-generation earth
system models.

**Code and Data Availability**

The MITgcm and user manual are available from the project website: http://mitgcm.org/. The ECCOv4
setup can be found at    http://wwwcvs.mitgcm.org/viewvc/MITgcm/MITgcm_contrib/llc_hires/. The
exact version of MITgcm, ECCOv4 configuration, MATLAB routines to process the ECCOv4 output,
generate the target model skill assessment diagram, and produce the paper figures are archived on Zenodo
(doi:10.5281/zenodo.4106405). The SMAP observations can be downloaded from
http://apdrc.soest.hawaii.edu/las/v6/dataset?catitem=2928. The model forcing and simulated salinity
fields at different resolutions are archived on Zenodo (doi:10.5281/zenodo.4095613)

**Author Contribution**

Dimitris Menemenlis designed the experiments and Hong Zhang carried them out. Yang Feng, Dimitris
Menemenlis, Dustin Carroll developed the model code and performed the simulations. Yang Feng
prepared the manuscript with contributions from Huijie Xue, Dustin Carrol, Yan Du, and Hui Wu

**Acknowledgment**

The work was supported by CAS Pioneer Hundred Talents Program Startup Fund (Y9SL11001);
Southern Marine Science and Engineering Guangdong Laboratory (Guangzhou) (GML2019ZD0303,
2019BT2H594), ISEE2018PY05 from Chinese Academy of Sciences; the Chinese Academy of
Sciences (XDA15020901; 133244KYSB20190031), and Guangdong Key Laboratory of Ocean Remote
Sensing (South China Sea Institute of Oceanology Chinese Academy of Sciences) (2017B030301005-
LORS2001). D.M., D.C., and H.Z., carried out research at Jet Propulsion Laboratory, California
Institute of Technology, under contract with NASA, with grants from Biological Diversity, Physical
Oceanography, and Modeling, Analysis, and Prediction Programs.




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
