# Peer review of "I Improved representation of river runoff in Estimating the"

_Geoscientific Model Development, 2020_

## Referee Comment (RC1) · Anonymous Referee #1 · 19 Nov 2020

This is the review of "Improved representation of river runoff in Estimating the Circulation and Climate of the Ocean Version 4 (ECCOv4) simulations: implementation, evaluation and impacts to coastal plume regions"by Feng et al.

This paper proposes a new representation of global river runoff (i.e., real freshwater flux of daily, Point-source Runoff, DPR) in the ECCOv4 framework, which allows for a more realistic treatment of coastal plume dynamics and thus obtains better simulation of SSS and plume properties, compared to independent observations. Also, parallel experiments were conducted to evaluate the impacts of model grid resolution and grid type. Target diagrams were introduced to compare simulation results with SMAP observation systematically. It found that increasing model resolution from the coarse resolution LLC90 grid to the intermediate resolution LLC270 grid elevates the river plume and improves the SSS simulation, but the impacts of further increasing the resolution is subtle and regionally dependent, likely due to specific regional bathymetry and dynamic environments. Grid types seem not controlling the results.

Land-Ocean Interface is a weakly simulated component in most global-scale models. Not only because the estuaries and watersheds are usually not included, but also the spatiotemporal riverine freshwater (plumes) cannot be well resolved. This work is a significant improvement to the ECCO syntheses. Considering ECCO products world-widely used, an improvement in plume simulation will advance studies in diagnosing fate of land source pollutants (e.g. Microplastic); closing the carbon budget estimation gap and et al.. . . I feel the proposed technology is significant and novel. The results, including the new representation of global river runoff, and the evaluations of the representation in different model setups (which are in very great detail), offer a benchmark for other global models. In fact, this representation seems very easy to be applied onto other models. This paper fits the GMD journal scope very well. The paper is well written, flows well from topic to topic, is clear and understandable. The figures are excellent. I would suggest Minor Revision to this version.

I hope the authors consider the following minor comments that I found need to be clarified or complemented.

L28: "Sea-Surface Salinity"-> "Sea Surface Salinity" L45: add a comma after "products' L54: "Sea Surface Salinity"-> "Sea Surface Salinity (SSS)" L113:"locally isotropic"-> please make sure if they are really isotropic. L118: "The model has 50 vertical z-levels; vertical resolution is 10 m in the. . .", the same for all the experiments that you run? Please clarify. L121: GGL should be defined. L127: "ECCOv4 uses natural boundary conditions" -> "ECCOv4 uses natural boundary conditions for the river discharge"? L143: Please provide details for the " iso-neutral mixing" and "residual mean velocity". L149: What are the initial conditions and surface forcings for the ECCOv4 here? The optimized or initial fields? L182: "in at"->"at", "white boxes"->"black boxes" L198: Is this equation applied to each grid point, or to an area? L217: Eq. 7 is not correct. Reformulate it or delete it- it does not influence the analysis. L237, L243: Is the integration in the formulas bounded by specific S? L243: What is the effect of vertical resolution? L297: "grid resolution"->"river forcing"? L301: Figures 5? Figs. 4&5. Please add discussion on the reasons of the large bias over 1.5? Fig. S4: why the seasonal variations are different between R and C experiments? L377-383: please explain EOF2. L386: reference should not be italic. L389: NBC should be defined in advance. Fig. 8. Caption. "Same as Fig. 7", not. L438: "experiments ability"->" experiments' ability"? L462, Fug11; what is "integrated freshwater transport"? Why does the transport of the 3 rivers show increasing trend over the 3 years ? Why no annual cycle for Amazon? Please explain. L484: 'stratification'-> 'stratification difference'? L591: VSF should be defined.

---

## Referee Comment (RC2) · Anonymous Referee #2 · 5 Jan 2021

The manuscript "Improved representation of river runoff in Estimating the Circulation and Climate of the Ocean Version 4 (ECCOv4) simulations: implementation, evaluation and impacts to coastal plume regions" by Feng et al. deals with an effort to improve the representation of river runoff in ECCOv4-based ocean simulations. For the purpose, the authors use both diffusive climatological runoff and daily point-source runoff to drive the river-related freshwater forcing in their simulations. Results from a series of experiments using different grid types and resolutions lead to detailed conclusions

about the agreement between simulated and remotely-sensed sea surface salinity, and about the impacts on river plume area and volume, freshwater thickness and transport and mixed layer depth in the low- to mid-latitude ocean.

The topic, of a technical nature, is interesting and has direct impact on the modelling community. The manuscript is well written and structured and the figures presented back up the conclusions reached. I suggest acceptance after a moderate revision taking into account the following points.

My first criticism deals with the comparisons with observations. On one hand, the synchronized validation against SMAP SSS is done only over a period of 33 months. On the other hand, there is this discrepancy between using the WOA18 (including data from 1955-2010) and a climatology from the simulation based on only years 2015-2017. Could you please comment on the significance of such comparisons? Maybe emphasize the caveats in the text...

My second major concern: If I understood correctly, the difference between C and R runs is not only the forcing method (single point source versus area adjacent to river mouth) but also the temporal resolution of the prescribed runoff (monthly in C vs daily in R) and actually the dataset that serves as basis. That complicates the comparisons in my opinion and one is not sure if the improvements between C and R at a given resolution are entirely attributable to the forcing methodology (the benefit of which, I believe, is the point the authors are trying to make) or, on the other hand, additionally due to temporal resolution in the forcing. Please elaborate on this problem.

My last main question is: Why is there no simulation including the climatological forcing at the LLC540 resolution? Would make the comparisons more robust. The authors should also try to explain why the finest resolution seems to present a poorer comparison to observations.

Below some minor comments/corrections the authors should take into account in their revised version:

L44-46: Please explain in which sense are the results a benchmark?

L243: Should it read -h instead of -H?

L306: Do you mean the negative bias is reduced?

L315-317: One sees only a slight tendency and mainly in the case of the Amazon river. I would be very careful in assuming that the interannual variability is reproduced, since it depends on many factors other than the prescribed runoff...

Figure 3: Why is LLC270C so much better than the other resolutions in the Amazon case?

Figure 5: What is the reason for the pronounced normalized bias increase from LLC270R to LLC540R?

Caption Figures S4 and S5: I believe you mean Columbia.

L443: "and responds".

L430: Caption of Figure 8 is wrong. I actually do not see the point of showing both area and volume. They do not differ much in the variability they present. I suggest presenting only volume.

L372: North Brazil Current and not Brazil Current.

L458: CO instead of CR.

Figure 10: Caption is incomplete.

L462: It is not clear how the transports were calculated. What do the authors mean by arc? I suggest completely removing the transport calculations from the discussion.

L540-543: I do not understand what is meant here. Increasing resolution always increases the richness of spatio-temporal scales!

---

## Author Comment (AC1) · 12 Feb 2021

Response-to-reviewer 1:

We thank reviewer 1's encouraging words and constructive comments to improve the work. Please check out our point-by-point response below. The line numbers below are correspond to the revised manuscript.

[Figure]

L28: "Sea-Surface Salinity"-> "Sea Surface Salinity" Response: Done. Please refer to line 28.

L45: add a comma after "products' Response: The statement has been removed after response to the second reviewer's comments, so the word no longer exist.

L54: "Sea Surface Salinity"-> "Sea Surface Salinity (SSS)" Response: Done. Please refer to Line 54.

L113:"locally isotropic"-> please make sure if they are really isotropic.

Response: We've checked with the ECCO community and found this and the the next statement may only hold for ECCOv4 LLC90. Please refer to Figure 1 in Forget et al. (2015) Geosci. Model Dev. https://doi.org/10.5194/gmd-8-3071-2015. We removed them accordingly.

L118: "The model has 50 vertical z-levels; vertical resolution is 10 m in the. . .", the same for all the experiments that you run? Please clarify. Response: Yes, all experiments have the same configuration for the vertical grid setup. We have clarified this in the text by adding the sentence, "This setup was the same for all designed experiments". Please refer to the revised manuscript Line 116.

L121: GGL should be defined. Response: This is short for Gaspar-Gregoris-Lefevre (Gaspar et al., 1990). Please refer to Line 118 to Line 119.

Gaspar, P., Grégoris, Y. & Lefevre, J.-M., A simple eddy kinetic energy model for simulations of the oceanic vertical mixing: Tests at station Papa and long-term upper ocean study site, 95(C9), 16179-16193, doi:10.1029/JC095iC09p16179, 1990

L127: "ECCOv4 uses natural boundary conditions" -> "ECCOv4 uses natural boundary conditions for the river discharge"? Response: Natural boundary condition is applied for both river discharge as well as Evaporation-Precipitation (E-P). Therefore, we changed it to "ECCOv4 used natural boundary conditions for both the river discharge and E-P (evaporation minus precipitation). Please refer to Line 125 to Line 126.

[Figure]

L143: Please provide details for the "iso-neutral mixing" and "residual mean velocity". Response: The details have been provided as "D_(v,S) and D_($\sigma$,S) are subgrid-scale processes parameterized as mixing diapycnal and along the isoneutral surface, which respect the highly adiabatic process of the oceanic interior (Griffies et al. 1998). The v_res and w_res are the horizontal and vertical residual mean velocity fields and hold the relationship (v_res,w_res)=(v,w)+(v_b,w_b), where (v_b,w_b) is the bolus velocity parameterize the effect of unresolved eddies (Gent and Mcwilliams, 1990)."

Please refer Line 139-143 for the details and two additional references

Gent, P. and Mcwilliams, J.: Isopycnal mixing in ocean circulation models, J. Phys. Oceanogr., 20, 150–155, 1990. Griffies, S. M., Gnanadesikan, A. Pacanowski, R. C., Larichev, V. D., Dukowicz, J. K. and Smith, R.D. Isoneutral Diffusion in a z-Coordinate Ocean Model, Journal of Physical Oceanography 28, 5 (1998): 805-830, doi:10.1175/1520-0485(1998)028<0805:IDIAZC>2.0.CO;2, 1998

L149: What are the initial conditions and surface forcings for the ECCOv4 here? The optimized or initial fields? Response: The initial condition of ECCOv4 was from optimized adjustment of Mapping Ocean Observations in a Dynamical Framework: A 2004-06 Ocean Atlas (OCAA) and surface forcing was from adjustment of ECMWF Re-Analysis (ERA) interim. Please see Line 149-Line 152 in the revised manuscript

L182: "in at"->"at", "white boxes"->"black boxes" Response: Done. Please refer to Line 189.

L198: Is this equation applied to each grid point, or to an area? Response: This calculation is applied to an area, which has been recognized as the river-month region in Line 265-272 to recognize the river mouth region. Data was averaged within the river mouth to generate time series and then do a skill calculation using this formula. We have further clarified this in Line 197.

L217: Eq. 7 is not correct. Reformulate it or delete it- it does not influence the analysis.

Response: We have removed Eq. 7.

L237, L243: Is the integration in the formulas bounded by specific S? Response: Yes. This was introduced in Section 4.2 in Line 447. A series of S has been used as the threshold (from 28 to 36 PSU) for the plume area calculation. S = 30 PSU was specified for the plume volume calculation. We also have switched Figure 8 and Figure S6 in response to reviewer 2's comments.

L243: What is the effect of vertical resolution? Response: Reviewer 2 request us remove the transport calculation. We have do so accordingly. Therefore, there was no this line in the revised manuscript. We'll leave more plume analysis in future works.

L297: "grid resolution"->"river forcing"? Response: We thank the reviewer read this carefully. This should be runoff forcing as well as the model grid resolution. We have changed accordingly. Please refer to Line 325 in the revised manuscript.

L301: Figures 5? Figs. 4&5. Please add discussion on the reasons of the large bias over 1.5? Response: We thank the reviewer check our manuscript carefully. This should be Fig. 4 rather than Fig. 5. We have fixed it. The Mekong river had large normalized bias over 1.5 for LLC270R and LLC540R. A plot SMAP SSS timeseries in the MK river mouth area (see attached Figure) shows that the low salinity signal associated with riverine freshwater has not been well recognized. It also had a lot of sub-montnly variabilities (noise). This may be because SMAP SSS are contaminated by land signals near the Vietnam coast. Therefore, taking the SMAP is abnormal comparing to other river mouth regions. We have added a comment from Line 369 to Line 371.

Fig. S4: why the seasonal variations are different between R and C experiments? Response: The R and C experiments are distinct from each other not only for the way freshwater added to multiple grids or one-single grid, but also for the river forcing itself. The R experiment used JRA55DO forcing, which had both seasonal and inter-annual variability. The C experiment used Fekete et al. (2002), which is climatological, with
seasonal variability only. The are not exactly the same. This was also questioned by reviewer 2. To further check on how much variability brought by the river forcing temporal variability itself, we run two additional experiments: Exp. LLC270R_spread, which used diffusive surface forcing method, but daily JRA55DO runoff. Exp. LLC270R_clim, which used point-source surface forcing, but climatological runoff derived from 2015-2017 JRA55DO (Table 1). We updated Table 1-3, Figure 3 and 4, and placed a new figure in supplementary material (S8) for the new experiments. We updated the corresponding statement from Line 163 to Line 171; Line 278 to Line 291; Line 307 to Line 316; Line 341 to Line 352; Line 360 to Line 371. We now refer to adding runoff to multiple cells from the surface as the diffusive runoff; to a single grid cell as the point-source runoff.

L377-383: please explain EOF2. Response: The spatial pattern of the second EOF mode represents the low salinity Mississippi River plume water transport downcoast from Louisiana towards Texas, which was carried by the reversed shelf circulation from September to May (Cochrane and Kelly, 1986). We also added the following references Cochrane, J. D., Kelly, F. J., 1986. Low-frequency circulation on the Texas-Louisiana continental shelf. J. Geophys. Res. 91(C9), 10645-10659. Please refer to Line 420 to Line 422 in the revised manuscript.

L386: reference should not be italic. Response: Done. See Line 425.

L389: NBC should be defined in advance. Fig. 8. Caption. "Same as Fig. 7", not. Response: Done, please refer to Line 437. Fig. 8 has been switched with Figure S6 in supplementary material.

L438: "experiments ability"->" experiments' ability"? Response: Done. Please refer to Line 478.

L462, Fug11; what is "integrated freshwater transport"? Why does the transport of the 3 rivers show increasing trend over the 3 years? Why no annual cycle for Amazon? Please explain.

Response: The integrated transport was calculated by taking the starting point as day 1; then day2 was the integration of day1 and day2 (day1 + day2); and day3 integrated day1, 2 and 3 (day 1 + day 2 + day 3). Reviewer 2 suggested we removed the freshwater transport part in this manuscript, we have done so accordingly. So, there was no longer this part in the main text. We'll leave more plume analysis to our future works.

L484: 'stratification'-> 'stratification difference'? Response: Done. Please refer to Line 511 in the revised manuscript.

L591: VSF should be defined. Response: Done. Please refer to Line 617 in the revised manuscript.

Please also note the supplement to this comment:
https://gmd.copernicus.org/preprints/gmd-2020-321/gmd-2020-321-AC1-supplement.pdf

―――――――――――――――――――

[Figure]

**Fig. 1.** SSS timeseries at the Mekong River from SMAP

---

## Author Comment (AC2) · 12 Feb 2021

Response-to-reviewer 2:

We thank reviewer 2's thoughtful suggestions, which help us to improve our work. The question regarding the difference between C and R runs is excellent. We have since designed two additional experiments, LLC270R_spread and LLC270R_clim, to address this question. In addition, we have responded other comments one-by-one as

follows. The line numbers below are referred to the revised manuscript.

My first criticism deals with the comparisons with observations. On one hand, the synchronized validation against SMAP SSS is done only over a period of 33 months. On the other hand, there is this discrepancy between using the WOA18 (including data from 1955-2010) and a climatology from the simulation based on only years 2015-2017. Could you please comment on the significance of such comparisons? Maybe emphasize the caveats in the text...

Response: We thank the reviewer for bringing up this problem. We, the co-authors, had a lot of discussions on this part as we developed this work. The improvement to EC-COv4 mainly focused on the plume representations, for which there isn't enough in-situ observational data available publicly. SMAP is satellite observation. Although satellite products have nice spatio-temporal coverage as the ECCOv4 SSS products, previous studies of Mississippi river plume and Bay of Bangle found that all satellite SSS observation had some bias comparing to in-situ World Ocean Database (See Fournier et al. 2016; 2017). Fournier et al. (2016; 2017) adjusted the satellite SSS values to compensate for the bias in their studies. In contrast, WOA18 is an objective analysis of in-situ observations from a period of "climate normal" years (1981-2010) (Zweng et al., 2019). We compared ECCOv4 output with the WOA data for the absolute values of SSS (Table 2), which complements the comparisons with the SMAP data for the spatio-temporal patterns of plumes (Figures 2 &3). In addition, the comparisons in Table 2 and 3 demonstrate not only which ECCO experiment is closer to the climatology "truth", but also how ECCO products compared to SMAP. For example, in the Amazon River plume region, SMAP SSS underestimated WOA SSS by about 5.2 PSU. We hope this kind of information can be helpful for researchers to make informative decisions when using ECCO or SMAP products to pursue their scientific questions. We've already discussed the limitation of WOA and SMAP in Lines 246 – Line 256 in the revised manuscript. To highlight the point above, we have added the following statement "We firstly found that the SMAP SSS is lower than the WOA18 SSS for large rivers. The underestimation

is more than 5 PSU for the Amazon region." Please refer Line 272-Line 274 for the main text. We also added some discussions from Line 198 to Line 306 in the revised manuscript.

References: Fournier, S., Lee, T. and Gierach, M. M., Seasonal and interannual variations of sea surface salinity associated with the Mississippi River plume observed by SMOS and Aquarius, Remote Sensing of Environment, 180, 431 – 439. doi:10.1016/j.rse.2016.02.050, 2016 Fournier, S., Vandemark, D., Gaultier, L., Lee, T., Jonsson, B., and Gierach, M. M. Interannual variation in offshore advection of Amazon-Orinoco plume waters: Observations, forcing mechanisms, and impacts. Journal of Geophysical Research: Oceans, 122, 8966–8982. doi:10.1002/2017JC013103, 2017 Zweng, M.M, Reagan, J.R., Seidov, D., Boyer, T.P., Locarnini, R.A., Garcia, H.E., Mishonov, A.V., Baranova, O.K., Weathers, K.W., Paver, C.R. and Smolyar, I.V. World Ocean Atlas 2018, Volume 2: Salinity. A. Mishonov, Technical Editor, NOAA Atlas NESDIS 82, 50pp., 2019

My second major concern: If I understood correctly, the difference between C and R runs is not only the forcing method (single point source versus area adjacent to river mouth) but also the temporal resolution of the prescribed runoff (monthly in C vs daily in R) and actually the dataset that serves as basis. That complicates the comparisons in my opinion and one is not sure if the improvements between C and R at a given resolution are entirely attributable to the forcing methodology (the benefit of which, I believe, is the point the authors are trying to make) or, on the other hand, additionally due to temporal resolution in the forcing. Please elaborate on this problem.

Response: This is a great question. We appreciated the reviewer's insightful thoughts on the design of our experiment design. It is true that the difference between C and R could be attributed firstly to the diffusive versus point-source runoff; and secondly to the river discharge file itself. To further explore the problem, we did another two experiments based on the LLC270 gird, which showed the best performance when taking SMAP as the observational reference. Exp. LLC270R_spread, which used

the diffusive surface forcing method, but daily JRA55DO runoff. Exp. LLC270R_clim, which used point-source surface forcing, but climatological runoff derived from 2015-2017 JRA55DO (Table 1). We updated Tables 1-3, Figures 3 and 4, and placed a new figure in the supplementary material (S8) for further experiments. We updated the corresponding statement from Line 163 to Line 171; Line 278 to Line 291; Line 307 to Line 316; Line 341 to Line 352; Line 360 to Line 371. We now refer to adding runoff to multiple cells from the surface as the diffusive runoff; to a single grid cell as the point-source runoff.

From the updated Table 2, with the diffusive surface forcing, LLC270R_spread driven by daily JRA55DO had lower salinity than LLC270C driven by Fekete. This is not a surprise, since the model automatically interpolates the river forcing file to the model grids. The Fekete river discharge file spreads spatially more than JRA55DO before the model interpolation (Figure S1). This means less freshwater is added to the top of the interested river mouth region, resulting in a relatively high salinity. Moreover, with the JRA55DO forcing, the sea surface salinity with the point-source forcing (LLC270R) was lower than the diffusive forcing (LLC270R_spread). We can think about adding river to single grid cell instead of multiple cells was equivalent to decreasing the inlet width in regional models, which results in an increase in the inflow velocity, thus more efficiently spreading riverine freshwater within our selected river mouth area (Table 2). Lastly, our new runs show that with point-source river forcing, the runoff using year-by-year JRA55DO and climatological JRA55DO produced inconsistent SSS changes for selected rivers. Specifically, in the experiments with climatological JRA55DO, CG, CJ and PA river plumes have higher SSS, while AZ, GB, MR, MK and CO river plumes have lower SSS.

When taking SMAP as the reference, the model skill shows that LLC270R_spread is better than LLC270C for most rivers, such as AZ, GB, MR, PA, and CO. This is not surprising since JRA55DO runoff includes both seasonal and interannual variability, while Fekete only changes seasonally. As a comparison, the skill of LLC270R_spread

is worse than LLC270R. This is because diffusive forcing had higher SSS than point-source forcing (Figure S8). When taking WOA as the reference, the model skill shows climatological forcing run (LLC270R_clim and LLC270C) had better skill than daily forcing runs (LLC270R_spread and LLC270R). This is not surprising since WOA is a climatological dataset.

The difference between time-averaged LLC270 runs and SMAP are presented in Figure S8. The SSS bias was reduced near the Amazon by switching diffusive surface forcing from Fekete to JRA55DO. The positive bias became negative after switching diffusive to the point source. This also happened to the Mississippi and Columbia River. The SSS bias change is consistent with the above discussion for Table 2 and Table 3. The SSS time series near the river mouth are shown in updated figure 3. The 2017 spring Amazon flood can be seen when forced by diffusive daily JRA55DO (LLC270R_spread), but not by diffusive climatological Fekete case (LLC270C). The Mississippi/Columbia River mouth region is different from the Amazon in that the annual cycle of LLC270R_spread is stronger than in LLC270C in all three years. This is because the seasonality of the Mississippi-Atchafalaya/Columbia River mouth has been oversmoothed in the climatological Fekete. The LLC270R_clim fluctuates in comparable with the LLC270R, except the annual extreme low SSS were comparable for the three simulated year.

Lastly, we updated the original figure 4 (target diagram) by comparing LLC270C and LLC270R to LLC270R_spread and LLC270R. The reason is that the target diagram takes SMAP as the observational reference. SMAP SSS include both seasonal and interannual variability, hence it is more meaningful to compare the two cases driven by river discharge with both seasonal and interannual signals. With the same daily JRA55DO, we found the normalized bias is lower in the experiment with point sources than in the experiment with diffusive sources in general when forced with daily JRA55DO. The change in normalized unbiased RMSD are largely negligible compared to the changes in normalized bias for most rivers, except CG and MK. Most unbiased

RMSD remains negative when switching the runoff forcing from climatological to daily for most regions. This implies that the variance of LLC simulations remains lower than SMAP observations despite of the runoff forcing changes. The exception of the Congo River was possibility because it is a near equator eastern boundary plume where freshwater transport distinguish from others, while the exception of MK was because SMAP are contaminated by the land signal near Vietnam coast that the SSS timeriers had a big noise.

Since we are interested in how the new DPR implementation is different from the widely used general ECCOv4 (forced by Fekete runoff), we did not change the discussion part for impacts on river plume properties, MLD, and strength of stratification.

My last main question is: Why is there no simulation including the climatological forcing at the LLC540 resolution? Would make the comparisons more robust? The authors should also try to explain why the finest resolution seems to present a poorer comparison to observations.

Response: Again, this is a great question. We have tested the climatological diffusive river forcing and daily point source river forcing on ECCO CS510 grid first, which has a spatial resolution about 19 km. After switching to the LLC grid, the LLC540 setup had comparable resolution with CS510 grid set up. We had found that grid type switch with closer resolution had minor impacts on our studied rivers. Therefore, we anticipate that LLC540C vs LLC540R would be close to CS510C vs. CS510R. We agree with the reviewer's last comment "increasing resolution always increasing the richness of spatio-temporal scales", which is true. In our study, high resolution runs brought results further from SMAP, but they might not be further from the real world. Note that SMAP itself is a satellite product with 1/4o resolution, while LLC540R SSS at 1/6° resolution contains more dynamic features. We are sorry that part of our discussions and a comparison with SMAP may confuse the reviewer about the resolution impact. We've emphasized that we use SMAP and WOA18 as "observational references", where our model-observation comparisons provide useful information on how SSS change between experiments rather than determine which experiment is closer to the real world" (Please see Line 253-Line 256 in the main text). For further clarification, we changed the title of the section to "Comparison with SMAP and WOA18"

Below some minor comments/corrections the authors should take into account in their revised version:

L44-46: Please explain in which sense are the results a benchmark? Response: The word benchmark may not be appropriate. We have rewritten the implication of the research in the abstract. Please see Line 41 to Line 45 in the revised manuscript.

L243: Should it read -h instead of -H? Response: We removed the transport calculation as suggested comments below. So, this formula no longer exists.

L306: Do you mean the negative bias is reduced? Response: This should be the positive bias since the SSS in LLC#C is higher than the SMAP, thus the difference is positive.

L315-317: One sees only a slight tendency and mainly in the case of the Amazon river. I would be very careful in assuming that the inter-annual variability is reproduced, since it depends on many factors other than the prescribed runoff... Figure 3: Why is LLC270C so much better than the other resolutions in the Amazon case? Figure 5: What is the reason for the pronounced normalized bias increase from LLC270R to LLC540R? Caption Figures S4 and S5: I believe you mean Columbia.

Response: We agreed with the statement "interannual variability is reproduced" is not accurate, therefore, we have reworded this sentence to "when using DPR forcing, the SSS differences associated with the interannual variations of river discharge can be better represented". Please refer to Line 351 to Line 352 in the revised manuscript. Figure 3 shows in the Amazon region, the LLC270C, the coarse resolution run is closer to SMAP than CS510C, the fine resolution run. The CS510C run (the ECCO2 products) used the Stammer et al. (2004) runoff, whereas LLC270C (the ECCOv4 products) used

Fekete et al. (2002) runoff. A comparison between the Stammer et al. (2004) runoff and Fekete et al. (2002) runoff for the Amazon Region is now placed in Supplementary materials (Figure S7). We can see that the Stammer runoff was more diffusive spatially and without seasonal variability, which could be the reason for the CS510C run being worse than the LLC270C run. We added the following statement "The intermediate resolution LLC270C run is better than the high resolution CS510C run for the Amazon region. This is because the Stammer et al. (2004) runoff used in CS510C is smoother spatially and lacks seasonal variability compared to the Fekete et al. (2002) runoff in LLC270C for this region (Figure S7)" from line 348 to line 352.

For pronounced bias in Figure 5, the normalized bias was calculated as: B=(M ÌĚ-(Ref) ÌĚ)/$\sigma$_Ref The observational reference is SMAP. Therefore, the standard deviation (denominator) does not change when comparing different runs. The pronounced bias is because the SSS difference in LLC540R and SMAP was larger than the difference between LLC270R and SMAP. For the Amazon region, SSS from LLC270R was less than 1 PSU lower than the SMAP, while LLC540R was roughly 3 PSU lower. This also happened to the Columbia, where the absolute value of bias decreased by about 0.3 PSU after switching to the LLC540R. We added the statement "which is consistent with the SSS reductions shown in Table 2 and relatively low W_skill shown in Table 3" from Line 379-Line 380. "In addition, SMAP may underestimate SSS near the river mouth (Fournier et al. 2017). Therefore, larger biases in high resolution run does not indicate the simulation deviates from the truth" from Line 304 to Line 306.

The captions in Figures S4 and S5 have been fixed.

L443: "and responds". Response: Done. Please refer to Line 483

L430: Caption of Figure 8 is wrong. I actually do not see the point of showing both area and volume. They do not differ much in the variability they present. I suggest presenting only volume. Response: We have fixed the caption of Figure 8. It may be a little repetitive to show both area and volume at the given threshold. Therefore,

we switched the plume area calculation at different thresholds in Supplementary with the Figure in the Text. The original Figure S6 is Figure 8 in the revised text, and the original Figure 8 is Figure S6 in the revised supplementary. We also changed the corresponding text, please refer Line 447 to Line 454 in the revised manuscript.

L372: North Brazil Current and not Brazil Current. Response: Done. Please refer to Line 409.

L458: CO instead of CR. Figure 10: Caption is incomplete. Response: CO was switched to CR, and the Caption of Figure 10 has been fixed. Please refer to Line 498, Line 499 and Line 501.

L462: It is not clear how the transports were calculated. What do the authors mean by arc? I suggest completely removing the transport calculations from the discussion. Response: We removed the transport calculation in the method part, and corresponding text in the abstract, result, and discussion.

L540-543: I do not understand what is meant here. Increasing resolution always increases the richness of spatial-temporal scales! Response: We thank the reviewer for pointing this out, which pushes us to rethink the implications of this part of the results. ECCO is not just a numerical model, but also a suite of global ocean data and assimilation products that can be downloaded by researchers worldwide every day. Therefore, we change the implication to "Recently increases in computational power allowed GODAS products such as ECCO, to provide model output at different resolutions, which supports regional studies using data analysis approaches or offline modeling methods (e.g., the Lagrangian method, Meng et al. 2020; Liang et al. 2019). Our results suggest that how high-resolution products should be used depends on the interested spatio-temporal dynamics as well as geomorphology characteristics of the studied region itself". Hopefully, this statement can help ECCO-data users make better decisions in their research. Please see Line 565 to Line 569 in the revised manuscript and abstract.

Please also note the supplement to this comment:
https://gmd.copernicus.org/preprints/gmd-2020-321/gmd-2020-321-AC2-
supplement.pdf

———————————————————

[Figure]

**Fig. 1.** Updated Figure 3 with Exp LLC270R_spread and LLC270R_clim

1.5
Bias
1
0.5
ubRMSD
0
-1.5    -1    -0.5    0    0.5    1    1.5
-0.5
-1
-1.5

| | | | |
|---|---|---|---|
| ● Amazon LLC270R_spread | ● Mississippi LLC270R_spread | ■ Amazon LLC270R | ■ Mississippi LLC270R |
| ● Congo LLC270R_spread | ● Parana LLC270R_spread | ■ Congo LLC270R | ■ Parana LLC270R |
| ● Changjiang LLC270R_spread | ● Mekong LLC270R_spread | ■ Changjiang LLC270R | ■ Mekong LLC270R |
| ● Brahamptura LLC270R_spread | ● Columbia LLC270R_spread | ■ Brahamptura LLC270R | ■ Columbia LLC270R |

**Fig. 2.** Updated Figure 4 with LLC270R_spread and LLC270R

[Figure]

**Fig. 3.** Comparison between ECCO2(Cube-sphere grid) Stammer and ECCOv4 (LLC grid) Fekete runoff forcing (Figure S7)

[Figure]

**Fig. 4.** Zoomed-in view of SSS difference between different LLC270 experiments and SMAP observations for large (Amazon, a–d), medium (Mississippi, e–h), and small (Columbia, i–l) rivers (Figure S8).

| # | Experiment Name | Grid Type | Runoff Forcing | Grid spacing |
|---|---|---|---|---|
| 1 | LLC90C | Lat-Lon-Cap | ECCOv4 Climatology | 55–110 km |
| 2 | LLC90R | Lat-Lon-Cap | JRA55-do | 55–110 km |
| 3 | LLC270C | Lat-Lon-Cap | ECCOv4 Climatology | 18–36 km |
| 4 | LLC270R | Lat-Lon-Cap | JRA55-do | 18–36 km |
| 5 | LLC270R_spread | Lat-Lon-Cap | JRA55-do | 18–36 km |
| 6 | LLC270R_clim | Lat-Lon-Cap | JRA55-do | 18–36 km |
| 7 | LLC540R | Lat-Lon-Cap | JRA55-do | 9–18 km |
| 8 | CS510C (Standard ECCO2) | Cube-sphere | ECCO2 Climatology | ~19 km |
| 9 | CS510R | Cube-sphere | JRA55-do | ~19 km |

**Table 1:** Summary of all experiments. The ECCOv4 and ECCO2 climatological runoff is derived from Fekete et al. 2002 and Stammer et al. 2004, respectively. A comparison of runoff forcing is shown in Figure S1.

**Fig. 5.** Updated Table 1

GMDD
| River Mouth | Abb. | Discharge (m³/yr) | WOA18 | SMAP | LLC 90C | LLC 90R | LLC 270C | LLC 270R_spread | LLC 270R | LLC 270R_clim | LLC 540R | CS 510C | CS 510R |
|---|---|---|---|---|---|---|---|---|---|---|---|---|---|
| Amazon/ Orinoco | AZ | 6440 | 32.7 | 27.5 | 34.0 | 34.1 | 31.7 | 30.4 | 28.2 | 27.6 | 24.6 | 34.3 | 23.8 |
| Congo | CG | 1270 | 33.6 | 33.7 | 34.7 | 34.3 | 34.6 | 35.3 | 33.9 | 35.2 | 33.7 | 34.9 | 34.1 |
| Changjiang | CJ | 907 | 32.9 | 31.4 | 33.1 | 32.8 | 33.0 | 33.2 | 32.2 | 32.4 | 31.8 | 32.5 | 30.9 |
| Ganges /Brahamptura | GB | 643 | 29.3 | 27.5 | 30.9 | 29.4 | 29.5 | 27.6 | 27.2 | 27.1 | 23.9 | 29.7 | 25.6 |
| Mississippi | MR | 552 | 33.5 | 34.8 | 35.8 | 34.7 | 35.8 | 34.4 | 34.1 | 33.3 | 33.8 | 35.3 | 34.1 |
| Parana | PA | 517 | 28.9 | 27.3 | 33.7 | 31.0 | 31.1 | 24.9 | 24.7 | 25.5 | 20.0 | 33.8 | 20.0 |
| Mekong | MK | 504 | 32.9 | 32.9 | 33.5 | 32.6 | 32.3 | 31.2 | 30.3 | 29.8 | 31.0 | 31.8 | 28.5 |
| Columbia | CO | 167 | 30.7 | 31.0 | 32.0 | 31.7 | 31.7 | 31.3 | 30.8 | 30.5 | 30.3 | 31.4 | 30.4 |

294 **Table 2:** The SSS near river mouth for WOA18, SMAP, and all experiments for the selected regions

**Fig. 6.** Updated Table 2

| River Mouth | SMAP | LLC 90C | LLC 90R | LLC 270C | LLC270R_ spread | LLC 270R | LLC270R_ clim | LLC 540R | CS 510C | CS 510R |
|---|---|---|---|---|---|---|---|---|---|---|
| with SMAP | | | | | | | | | | |
| Amazon / Orinoco | - | 0.50 | 0.50 | 0.71 | 0.83 | 0.92 | 0.92 | 0.79 | 0.46 | 0.73 |
| Congo | - | 0.58 | 0.64 | 0.69 | 0.46 | 0.89 | 0.47 | 0.88 | 0.60 | 0.87 |
| Changjiang | - | 0.53 | 0.59 | 0.51 | 0.50 | 0.64 | 0.31 | 0.83 | 0.59 | 0.85 |
| Ganges / Brahmaputra | | 0.61 | 0.71 | 0.69 | 0.83 | 0.85 | 0.84 | 0.69 | 0.57 | 0.70 |
| Mississippi | | 0.55 | 0.79 | 0.53 | 0.75 | 0.77 | 0.69 | 0.75 | 0.49 | 0.72 |
| Parana | | 0.37 | 0.51 | 0.45 | 0.60 | 0.62 | 0.56 | 0.40 | 0.37 | 0.40 |
| Mekong | | 0.79 | 0.90 | 0.77 | 0.67 | 0.54 | 0.47 | 0.63 | 0.74 | 0.38 |
| Columbia | | 0.46 | 0.60 | 0.49 | 0.68 | 0.73 | 0.70 | 0.74 | 0.27 | 0.61 |
| with WOA | | | | | | | | | | |
| Amazon / Orinoco | 0.47 | 0.73 | 0.69 | 0.87 | 0.62 | 0.64 | 0.78 | 0.47 | 0.54 | 0.44 |
| Congo | 0.54 | 0.60 | 0.67 | 0.70 | 0.48 | 0.94 | 0.47 | 0.95 | 0.64 | 0.92 |
| Changjiang | 0.44 | 0.82 | 0.94 | 0.68 | 0.70 | 0.70 | 0.73 | 0.72 | 0.78 | 0.54 |
| Ganges / Brahamptura | 0.73 | 0.72 | 0.90 | 0.92 | 0.77 | 0.78 | 0.82 | 0.51 | 0.73 | 0.59 |
| Mississippi | 0.69 | 0.46 | 0.59 | 0.46 | 0.76 | 0.68 | 0.60 | 0.73 | 0.49 | 0.66 |
| Parana | 0.87 | 0.40 | 0.45 | 0.45 | 0.42 | 0.42 | 0.42 | 0.29 | 0.40 | 0.29 |
| Mekong | 0.64 | 0.84 | 0.87 | 0.83 | 0.39 | 0.45 | 0.59 | 0.54 | 0.71 | 0.30 |
| Columbia | 0.47 | 0.51 | 0.62 | 0.63 | 0.85 | 0.87 | 0.82 | 0.84 | 0.51 | 0.90 |

**Table 3:** The Willmott skill score for each run as compared with WOA18 and SMAP. The river mouth was recognized by the 1st EOF of WOA18 (See Figures 5 and S1). Note that WOA18 data are a 30- year climatology (1981—2010) and not in the same period as SMAP and experiments.

**Fig. 7.** Updated Table 3